# Object-Centric Representation Learning for Enhanced 3D Semantic Scene Graph Prediction

**KunHo Heo***
Kyung Hee University
hkh7710@khu.ac.kr

**GiHyun Kim***
Kyung Hee University
kimh060612@khu.ac.kr

**SuYeon Kim**
Kyung Hee University
spoiuy3@khu.ac.kr

**MyeongAh Cho**†
Kyung Hee University
maycho@khu.ac.kr

## Abstract

3D Semantic Scene Graph Prediction aims to detect objects and their semantic relationships in 3D scenes, and has emerged as a crucial technology for robotics and AR/VR applications. While previous research has addressed dataset limitations and explored various approaches including Open-Vocabulary settings, they frequently fail to optimize the representational capacity of object and relationship features, showing excessive reliance on Graph Neural Networks despite insufficient discriminative capability. In this work, we demonstrate through extensive analysis that the quality of object features plays a critical role in determining overall scene graph accuracy. To address this challenge, we design a highly discriminative object feature encoder and employ a contrastive pretraining strategy that decouples object representation learning from the scene graph prediction. This design not only enhances object classification accuracy but also yields direct improvements in relationship prediction. Notably, when plugging in our pretrained encoder into existing frameworks, we observe substantial performance improvements across all evaluation metrics. Additionally, whereas existing approaches have not fully exploited the integration of relationship information, we effectively combine both geometric and semantic features to achieve superior relationship prediction. Comprehensive experiments on the 3DSSG dataset demonstrate that our approach significantly outperforms previous state-of-the-art methods. Our code is publicly available at https://github.com/VisualScienceLab-KHU/OCRL-3DSSG-Codes.

## 1 Introduction

Recent research in 3D Semantic Scene Graph (3DSSG) Prediction has significantly enhanced the semantic understanding of 3D environments. By abstracting raw point cloud data into structured semantic graphs that capture objects and their interrelationships, 3DSSG facilitates critical tasks such as robot navigation [18, 57, 17], object manipulation [11], and VR/AR interactions [42]. This technology has become essential for semantic-level 3D understanding in various applications [40, 2, 33, 4, 12] by reducing ambiguity and facilitating intuitive human-machine interaction.

A variety of deep learning approaches have been proposed for accurate 3DSSG prediction [3, 24, 15, 38, 46, 51, 61, 13]. SGPN [45] pioneered the point cloud-based relationship prediction framework and released the 3DSSG dataset, which we utilize in this study. SGFN [52] introduced a feature-wise attention layer for more accurate inference and proposed an incremental update methodology. Most

---

* Equal contribution
† Corresponding author

39th Conference on Neural Information Processing Systems (NeurIPS 2025).

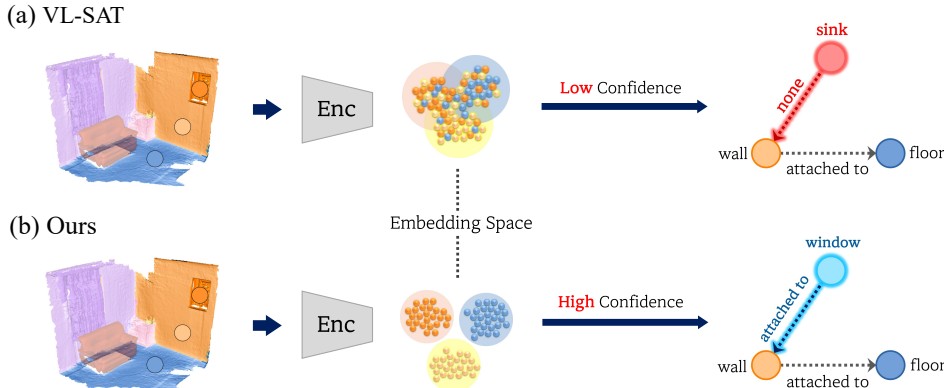

Figure 1: (a) VL-SAT [48] embeds object features non-discriminatively, leading to low-confidence predictions and frequent object misclassifications, which degrade relationship accuracy. In contrast, (b) our method embeds object features in a more discriminative manner, yielding high confidence scores and more accurate object classifications. Consequently, relationship predictions are significantly improved, resulting in a more coherent and semantically accurate scene graph.

recently, VL-SAT [48] addressed the problem of long-tailed distribution utilizing visual and textual information. Despite these advances, we identify two critical limitations in current approaches.

**Object misclassification leads to relationship errors.** Our analysis reveals that inaccuracies in object classification often propagate to relationship prediction, even in state-of-the-art methods. This is largely due to the limited discriminative power of object features, as many existing models prioritize relational inference using Graph Neural Networks (GNNs) without first ensuring robust object representation. As shown in Fig. 1(a), non-discriminative object embeddings result in low-confidence predictions and frequent misclassifications, ultimately degrading relationship accuracy.

To address this, we introduce a Discriminative Object Feature Encoder, pretrained separately to avoid entanglement with scene graph objectives. This encoder yields semantically rich and well-separated object embeddings, as shown in Fig. 1(b), serving as a reliable foundation for downstream graph construction. By leveraging these highly discriminative object features, our relationship feature encoder can effectively fuse semantic object identity with geometric cues, leading to more accurate relationship reasoning. This combined approach ensures that relationship prediction benefits from improved object classification that inherently provides richer semantic context, overcoming limitations of methods that rely primarily on geometric relationships.

**Lack of elaborating relationship information.** Prior works [50, 43, 55, 41, 29, 30, 58, 64] often fail to effectively integrate object information when constructing relationship features for prediction. Methods like SGFN [52], VL-SAT [48], and 3D-VLAP [47] rely solely on geometric relationship information between objects as edge features, neglecting object semantic characteristics. Conversely, approaches like Zhang et al. [63] and SGPN [45] use PointNet-based features from scene-level point clouds containing both objects, introducing excessive background information that can degrade prediction accuracy.

We propose a novel Relationship Feature Encoder that jointly embeds object pair representations with explicit geometric relationship information. Given the inherent disparity in dimensionality and information content, we introduce a Local Spatial Enhancement (LSE) module that applies targeted regularization. This encourages preserving geometric metadata while maintaining the integrity of the feature of the object, effectively mitigating the representational imbalance. Additionally, we design GNN with Bidirectional Edge Gating (BEG), enabling separate encoding of subject and object roles based on edge directionality. This directional decomposition explicitly captures asymmetric relational semantics, which are often overlooked in prior symmetric modeling. We further incorporate Global Spatial Enhancement (GSE) that contextualizes object relationships by integrating holistic geometric placement information, allowing the model to capture global spatial dependencies for accurate relationship prediction.

Our main contributions are as follows: (1) We identify the overlooked importance of object representation in prior 3DSSG methods and propose a **Discriminative Object Feature Encoder**, pretrained

independently to serve as a robust semantic foundation—improving not only our model but also enhancing performance when integrated into existing frameworks; (2) A novel **Relationship Feature Encoder** that combines object pair embeddings with geometric relationship information, enhanced by LSE; (3) A **Bidirectional Edge Gating** mechanism that explicitly models subject-object asymmetry, along with a **Global Spatial Enhancement** to incorporate holistic spatial context; We validate our approach through extensive experiments, achieving significant performance improvements over state-of-the-art 3DSSG methods.

## 2 Observations

A first inspection of validation scenes reveals that relation mistakes rarely occur in isolation: when the model assigns an incorrect label to either the subject or the object, the accompanying predicate, which describes the relationship between them, is also likely to be wrong. To quantify this phenomenon, we categorize all predictions into three mutually-exclusive groups according to the correctness of the object labels: (1) *Correct Object / Correct Subject*, (2) *Wrong Object / Correct Subject  or  Correct Object / Wrong Subject*, and (3) *Wrong Object / Wrong Subject*.

Table 1 reports the distribution of these groups for the previous models (SGPN [45], SGFN [52], and VL-SAT [48]). Only $8\%$ of VL-SAT's predicate errors occur when both objects and subjects are correctly recognized, whereas the misclassification ratio increases significantly in the remaining two categories. Other works like SGPN and SGFN also follow this trend, suggesting that improving object classification performance is a promising approach for reducing predicate errors as well.

| **Model** | **Obj. ✓ Sub. ✓** | **Obj. ✓/✗ Sub. ✗/✓** | **Obj. ✗ Sub. ✗** |
|---|---|---|---|
| SGPN [45] | 8% | 12% | 18% |
| SGFN [52] | 8% | 12% | 20% |
| VL-SAT [48] | 8% | 13% | 19% |

Table 1: Distribution of **predicate errors** according to object and subject classification status (✓ : correct classification, ✗ : misclassification).

Furthermore, as shown in Fig. 2, the predicate prediction error increases almost monotonically with the entropy $H(o|\mathbf{z})$ of the object classification, where $o$ is the object label predicted from the feature vector $\mathbf{z}$. This trend holds even when both objects are correctly predicted at Top-1 accuracy, and is consistent across entropy bins with similar relationship frequencies. This result guides our work toward designing a *more discriminative and accurate object feature space* for improved predicate estimation. Based on these observations, we hypothesize in the following section that predicate classification errors are strongly correlated with incorrect or uncertain object label predictions (please refer to the Appendix Section B for more details).

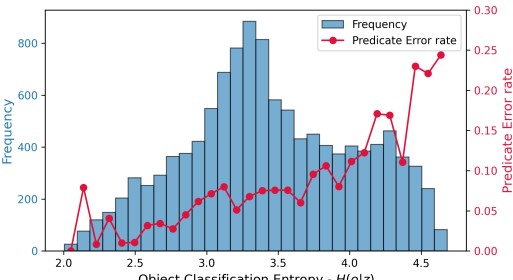

Figure 2: Histogram of object classification entropy and predicate prediction error rate, illustrating that higher entropy is associated with increased predicate errors under comparable relationship frequencies.

**Probabilistic formulation.** Let $o_i, o_j \in \mathcal{O}$ denote the ground-truth semantic labels of two objects, $e_{ij} \in \mathcal{E}$ the predicate connecting them, and $\mathbf{z}_i, \mathbf{z}_j$ the embedding vectors produced by an object encoder $f_{\theta_p}$. The probability of correctly classifying an object given its embedding $\mathbf{z}_i$ is $P(o_i \mid \mathbf{z}_i)$, while predicate classification is governed by the conditional distribution $P(e_{ij} \mid \mathbf{z}_i, \mathbf{z}_j)$. When the predicate head relies—either explicitly or implicitly—on object semantics and confidence of estimation, we can approximate $P(e_{ij} \mid \mathbf{z}_i, \mathbf{z}_j) \approx P(e_{ij} \mid o_i, o_j)$, which yields the factorization

$$P(e_{ij} \mid \mathbf{z}_i, \mathbf{z}_j) = \sum_{o_i', o_j' \in \mathcal{O}} P(e_{ij} \mid o_i', o_j') \, P(o_i' \mid \mathbf{z}_i) \, P(o_j' \mid \mathbf{z}_j). \tag{1}$$

Eq. (1) makes explicit that *sharper* object posteriors $P(o \mid \mathbf{z})$—i.e., more discriminative embeddings—yield lower-entropy mixture and thus higher confidence in predicate prediction. In contrast, ambiguous object embeddings result in broader $P(o \mid \mathbf{z})$ distributions, diluting the contribution of correct object labels and increasing the likelihood of predicate errors. Similarly, prior studies in vision tasks—such as [32, 14, 9]—have demonstrated that incorporating prior object knowledge aids relation prediction. Notably, [5] leveraged the statistical co-occurrence of objects and explicitly

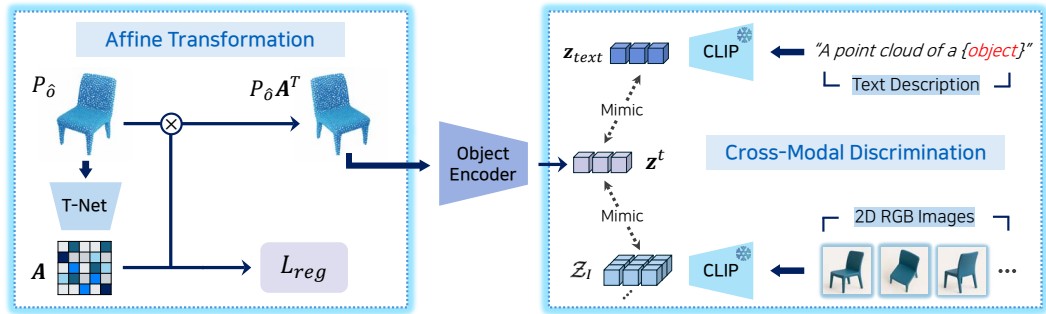

Figure 3: **Architecture of our Object Feature Encoder.** The encoder extracts object embedding $\mathbf{z}^t$ from point clouds via affine transformation, aligned with CLIP features: $\mathbf{z}_{\text{text}}$ from text description and $\mathcal{Z}_I$, a set of image features from multiple 2D RGB images.

regularized the semantic relationship prediction to address the uneven distribution issue. Compared to other works, we elaborated these assumptions with discriminative object features. For a more detailed comparison and analysis of our method, please refer to Section B of the Appendix.

Motivated by this analysis, we propose in Section 3.1 a contrastively pretrained encoder that maximizes the alignment between object embeddings and their class semantics. This sharpens the $P(o \mid \mathbf{z})$ and, through Eq. (1), improves both accuracy of predicate and triplet inference. The empirical results in Tables 2, 3 are fully consistent with this observations, providing evidence that improving object representation leads to more accurate and semantically consistent relation reasoning.

## 3 Proposed Methods

Our goal is to predict a 3D semantic scene graph represented as a directed graph $\mathcal{G} = \{\mathcal{V}, \mathcal{E}\}$ where $\mathcal{V}$ and $\mathcal{E}$ denote the sets of nodes and edges, respectively. The input consists of a 3D point cloud $\mathbf{P} \in \mathbb{R}^{N \times 3}$, containing $N$ points, a set of class-agnostic instance masks $\mathcal{M} = \{M_k\}_{k=1}^K$, which associate the point cloud with $K$ semantic instances. Also, 3DSSG takes the scan data of 3RScan [44] which contains RGB-D sequence when constructing the 3D point cloud data. Here, we can extract a collection of RGB images $\mathcal{I}_k = \{I_1, I_2, ..., I_{n_k}\}$ for each object instance in $\hat{o}_k \in \mathcal{V}$ following the procedure in VL-SAT [48]. Each object instance $\hat{o}_i$ represents the nodes in scene graph with ground-truth label $o_i \in \mathcal{O}$, while each edge $e_{ij} \in \mathcal{E}$ encodes the predicates in a triplet $\langle$ *subject, predicate, object* $\rangle$, where the head node $\hat{o}_i$ serves as the *subject* and the tail node $\hat{o}_j$ as the *object*. Each $\hat{o}_i$ corresponds to one of $N_{obj}$ semantic object classes, while each edge $e_{ij}$ can contain multiple predicate labels from $N_{pred}$ semantic relation classes such as 'standing on', 'hanging in', and others.

### 3.1 Object Feature Learning

As discussed in Section 2, learning discriminative object representations is crucial for accurate 3D semantic scene graph prediction. To this end, we pretrain the object encoder to provide the graph prediction model with robust semantic priors. Our object feature encoder (Fig. 3) employs a contrastive pretraining framework that leverages the correspondences between 3D object instances and their associated 2D image views, as well as between 3D object instances and their textual descriptions to enhance semantic expressiveness while preserving geometric invariance.

**Regularization for affine invariance.** To ensure robust 3D representations, our encoder must be invariant to affine transformations. As shown in Fig. 3, we adopt the T-Net architecture from PointNet [35], which predicts an affine transformation matrix $\mathbf{A} = \mathcal{T}(\mathbf{P}_{\hat{o}})$ for each object point cloud $\mathbf{P}_{\hat{o}}$. After applying this transformation, the object-level feature is extracted as $\mathbf{z}^t = f_{\theta_p}(\mathbf{P}_{\hat{o}}\mathbf{A}^T)$. To enforce orthogonality of the transformation matrix, we apply the following regularization term: $\mathcal{L}_{reg} = ||I - \mathbf{A}\mathbf{A}^T||_F^2$

**Discriminative object features with image/text modals.** As noted in Section 2, our objective is to pre-train an encoder that extracts discriminative object features, thereby enabling more confident object classification. To this end, we employ a contrastive pre-training scheme inspired by CrossPoint [1] and CLIP[2] [60], leveraging the text descriptions and 2D RGB images available for each 3D object instance. For an instance $\hat{o}_i$ with RGB views $\mathcal{I}_i = \{I_n\}_{n=1}^{n_i}$, we cast the task as a contrastive

learning problem with multiple positive samples per anchor. Features from both modalities are obtained with a pretrained CLIP [37] *ViT-B/32* encoder $f_{\theta_c}(\cdot)$. Specifically, each image is encoded as $\mathbf{z}^i_{\text{image}} \in \mathcal{Z}^i_I = \{f_{\theta_c}(I) : I \in \mathcal{I}_i\}$, while the textual description $T_{\hat{o}_i}$ is encoded as $\mathbf{z}^i_{\text{text}} = f_{\theta_c}(T_{\hat{o}_i})$. The text prompt follows the template *"A point cloud of {**object**}."*

Our objective is to maximize the similarity between the object embedding vector $\mathbf{z}^t$ and its corresponding image and text features $\mathbf{z}^i_{\text{image}}, \mathbf{z}^i_{\text{text}}$, while minimizing similarity to other embeddings in the mini-batch. Following previous works [56, 25, 60], we adopt a supervised contrastive framework to efficiently utilize label data, assuming that objects with the same label share semantic characteristics across their point clouds and associated images. However, unlike text prompts which provide a single representation per object, multiple images can correspond to a single object instance. To handle this asymmetry, we separate the contrastive losses for visual and textual modality. Let $\mathcal{D}_B = \{(\mathbf{z}^t_i, o_i)\}^B_{i=1}$ denote the batch of $B$ point cloud features and their corresponding ground-truth labels. For each index $i \in \mathcal{B} = \{1, ..., B\}$, we define the set of positive indices as $\mathcal{P}(i) = \{p \in \mathcal{B} : o_p = o_i\}$, and the set of negatives as $\mathcal{N}(i) = \mathcal{B}/\mathcal{P}(i)$. The visual contrastive loss is defined as:

$$\mathcal{L}^{visual}_i = \frac{1}{|\mathcal{P}(i)|} \sum_{p \in \mathcal{P}(i)} \sum_{\mathbf{z}_+ \in \mathcal{Z}^p_I} -\log \frac{\exp(s(\mathbf{z}^t_i, \mathbf{z}_+)/\tau)}{\sum_{r \in \mathcal{N}(i)} \sum_{\mathbf{z}_- \in \mathcal{Z}^r_I} \exp(s(\mathbf{z}^t_i, \mathbf{z}_-)/\tau)} \tag{2}$$

where $s(\cdot, \cdot)$ denotes the cosine similarity, and the temperature parameter $\tau$ is set to 0.07 during pretraining. Similarly, the text contrastive loss is defined as:

$$\mathcal{L}^{text}_i = -\log \frac{e^{s(\mathbf{z}^t_i, \mathbf{z}^i_{\text{text}})/\tau}}{\sum_{r \in \mathcal{N}(i)} e^{s(\mathbf{z}^t_i, \mathbf{z}^r_{\text{text}})/\tau}} \tag{3}$$

We compute the total cross-modal loss by averaging the individual losses across the batch:

$$\mathcal{L}_{cross} = \frac{1}{B} \left( \sum_{i \in I} \mathcal{L}^{visual}_i + \mathcal{L}^{text}_i \right) \tag{4}$$

Unlike prior works [1, 60], we do not include the positive samples in the denominator, thereby promoting a more robust and discriminative feature space. This design promotes the pretrained encoder to extract more discriminative features, which reduces conditional uncertainty $H(o \mid \mathbf{z})$ and leads to lower predicate prediction errors (please refer to the Appendix Section B for more details).

**Loss functions of object feature learning.** The training objective of the Object Feature Learning (OFL) is defined as $\mathcal{L}_{pretrain} = \lambda_{reg}\mathcal{L}_{reg} + \lambda_{cross}\mathcal{L}_{cross}$, where $\lambda_{cross} = 1$ and $\lambda_{reg} = 0.001$ are empirically set to balance the contributions of each term. This module is trained independently during a pretraining stage, after which the resulting encoder is employed as a fixed feature extractor in our scene graph prediction pipeline. This pretraining allows the object encoder to capture more discriminative and semantically meaningful representations.

## 3.2 Relationship Feature Learning

After the pretraining stage, we train the relationship feature encoder and the GNNs for scene graph prediction. Unlike prior approaches, our relationship feature encoder integrates both object-level semantic features and their geometric relationships. Following the design of geometric descriptors used in prior works [48, 52], we adopt their formulation to represent the spatial configuration between object pairs. The geometric descriptor between objects is defined as:

$$\mathbf{g}_{ij} = \text{CAT}\left( \boldsymbol{\mu}_i - \boldsymbol{\mu}_j, \boldsymbol{\sigma}_i - \boldsymbol{\sigma}_j, \mathbf{b}_i - \mathbf{b}_j, \log\frac{v_i}{v_j}, \log\frac{l_i}{l_j} \right) \in \mathbb{R}^{11} \tag{5}$$

where $\boldsymbol{\mu}$ and $\boldsymbol{\sigma}$ denote the mean and standard deviation of 3D point coordinates, $\mathbf{b} = (b_x, b_y, b_z)$ is the size of bounding box, and $v$ and $l$ represent the volume and maximum side length of the bounding box, respectively. Since many relationships are inherently related to spatial configurations, combining geometric descriptors with object features allows our encoder to capture both semantic and spatial cues. Our relationship feature encoder takes three inputs: the features of the subject and object (as defined in Section 3.1) and the geometric descriptor $\mathbf{g}_{ij}$. Given subject and object features $\mathbf{z}^t_i$ and $\mathbf{z}^t_j$, the initial edge feature is computed as $\mathbf{z}^e_{ij} = f_{\theta_r}\left( \text{CAT}\left( g_{\theta_{obj}}(\mathbf{z}^t_i), g_{\theta_{obj}}(\mathbf{z}^t_j), g_{\theta_{geo}}(\mathbf{g}_{ij}) \right) \right)$, where

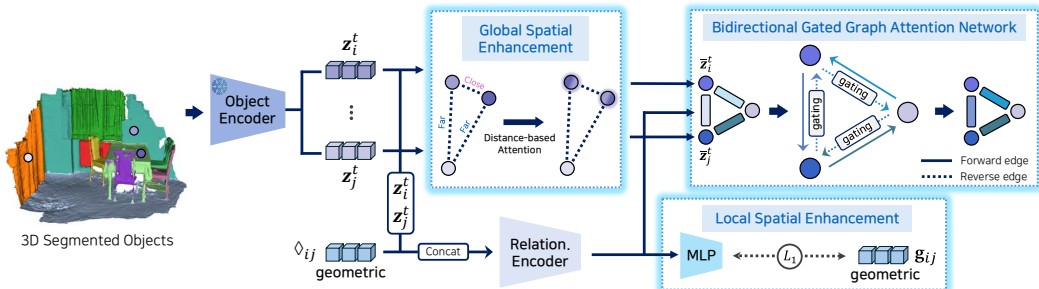

Figure 4: **Architecture overview.** Object embeddings $\{\mathbf{z}_i^t, \ldots, \mathbf{z}_j^t\}$ are refined via Global Enhancement to incorporate global spatial context based on inter-object distances, producing enhanced features $\{\bar{\mathbf{z}}_i^t, \ldots, \bar{\mathbf{z}}_j^t\}$. Simultaneously, the Local Spatial Enhancement locally preserves geometric relationships between object pairs. The Bidirectional Gated Graph Attention Network then selectively modulates the information of reverse edges, effectively capturing asymmetric relationships between objects.

$g_{\theta_{obj}}$ and $g_{\theta_{geo}}$ are MLP-based projection networks for object features and geometric descriptors, respectively. The function $f_{\theta_r}$ is a relation feature extractor implemented as a lightweight MLP followed by a 1D convolutional layer with kernel size 5. The operator $\text{CAT}(\cdot)$ denotes channel-wise feature concatenation.

To address the potential information imbalance between high-dimensional object embeddings and the relatively simple geometric descriptor, we introduce an auxiliary task, termed Local Spatial Enhancement (LSE), for the relationship feature extractor. This task employs an additional MLP-based projection head that aims to reconstruct the original geometric descriptor from the learned relationship feature. The objective is to minimize the $L_1$ loss between the predicted and original geometric descriptors. This auxiliary supervision encourages the relationship representation to retain geometric information, effectively compensating for the dimensional disparity between the object and geometric inputs.

### 3.3 Graph Neural Networks

**Global Spatial Enhancement.** In 3D scenes, spatial proximity and orientation between objects play a crucial role in determining their potential relationships. To incorporate this spatial awareness into our model, we adopted a Global Spatial Enhancement (GSE) mechanism. Given center coordinates $\boldsymbol{\mu}_i, \boldsymbol{\mu}_j$ of each node, we compute the Euclidean distance $d_{ij} = \|\boldsymbol{\mu}_i - \boldsymbol{\mu}_j\|_2$ between object pairs, explicitly capturing spatial relationships. With distance $d_{ij}$ between object instance pair $(i, j)$, we can represent the distances of all instance pairs as a matrix $D = [d_{ij}]_{i,j=1,\ldots,N}$. This distance matrix is multiplied by learnable parameters $W^{(h)} \in \mathbb{R}^{N \times N}$ to generate distance weights $w_{ij}^{(h)} = W^{(h)}D$. The computed distance weights are then integrated into the standard attention mechanism as follows:

$$\alpha_{ij}^{(h)} = \text{softmax}_j \left( \frac{\mathbf{q}_i^{(h)\top} \mathbf{k}_j^{(h)}}{\sqrt{d_k}} + \mathbf{w}_{ij}^{(h)} \right) \tag{6}$$

This formulation naturally reorganizes the spatial cues via $d_{ij}$ of objects within the scene, emphasizing geometrically meaningful relationships while effectively filtering object pairs with low relevance. Attention between spatially proximate objects can be strengthened, or object pairs with specific distance patterns can be emphasized, enabling the model to better understand the structural characteristics of the scene.

**Bidirectional Edge Gating.** Real-world object relationships exhibit inherent directionality, with distinct semantic roles assigned to subjects and objects. To capture this asymmetry, we propose a Bidirectional Edge Gating (BEG) mechanism that regulates information flow between directed edges. Throughout this process, we utilize node features $\bar{\mathbf{z}}$ that have been refined by the GSE module. To update node features, we separately aggregate outgoing and incoming edge features based on their respective roles:

$$\mathbf{z}_i^{\text{sub}} = \frac{1}{|E_i^{\text{sub}}|} \sum_{(i,j) \in E_i^{\text{sub}}} \mathbf{z}_{ij}^e, \quad \mathbf{z}_i^{\text{obj}} = \frac{1}{|E_i^{\text{obj}}|} \sum_{(j,i) \in E_i^{\text{obj}}} \mathbf{z}_{ji}^e \tag{7}$$

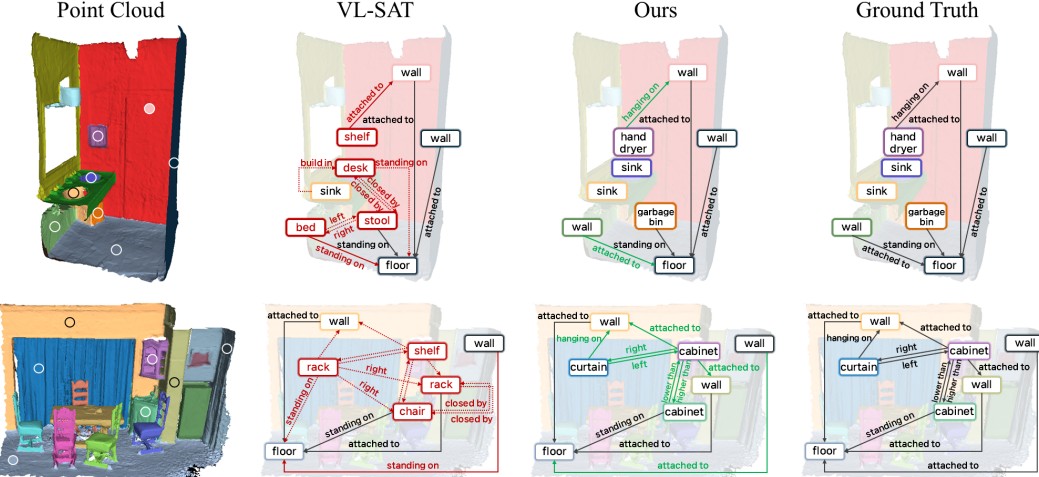

Figure 5: **3D scene graph visualizations.** → indicates true positive relations that are correctly predicted. → denotes false positives, where the model predicts an incorrect predicate for an existing relation. --→ represents either false negatives—missed ground-truth relations—or hallucinated relations that do not exist in the ground truth.

where $E_i^{\text{sub}}$ denotes the set of edges for which node $i$ serves as the subject, $E_i^{\text{obj}}$ denotes those where it serves as the object, and $\mathbf{z}_{ij}^e$ represents the edge feature from node $i$ to node $j$. Based on these updated subject and object feature of relation in graph, BEG updates node and directed edge feature $(i, j)$ as:

$$
\begin{aligned}
\bar{\mathbf{z}}_i &\leftarrow \text{LN}\Big(\text{MLP}\big(\bar{\mathbf{z}}_i,\ \sigma\big(W_{\text{dir}}\ \text{CAT}(\mathbf{z}_i^{\text{sub}},\ \mathbf{z}_i^{\text{obj}}))\big)\Big) \\
\mathbf{z}_{ij}^e &\leftarrow \text{MLP}\Big(\text{CAT}(\bar{\mathbf{z}}_i,\ \mathbf{z}_{ij}^e,\ \beta_{ij}\mathbf{z}_{ji}^e,\ \bar{\mathbf{z}}_j)\Big)
\end{aligned}
\tag{8}
$$

where $W_{\text{dir}}$ is learnable parameter, $\sigma(\cdot)$ is ReLU activation function, and LN, MLP are LayerNorm and MLP layer respectively. Additionally, while updating edge features in GNN, BEG controls the influence of reverse edge $(j, i)$ through a gate scalar $\beta_{ij} = \text{gate}(\mathbf{z}_{ij}^e)$. This bidirectional design preserves the semantic distinction between subject and object roles while enabling controlled information exchange across directed edges, thereby enhancing the model's ability to capture asymmetric relationships in 3D scene graphs.

### 3.4 Loss Functions

The training objective of our network is defined as $\mathcal{L}_{sg} = \lambda_{obj}\mathcal{L}_{obj} + \lambda_{rel}\mathcal{L}_{rel} + \lambda_{lse}\mathcal{L}_{lse}$, where $\mathcal{L}_{obj}$ is the object classification loss, computed using cross-entropy over object categories. The relationship classification loss, denoted as $\mathcal{L}_{rel}$, is calculated using binary cross-entropy. The third term, $\mathcal{L}_{lse}$, is the $L_1$ loss between the predicted and original geometric descriptors in the LSE task. The coefficients $\lambda_{obj}$, $\lambda_{rel}$, and $\lambda_{lse}$ control the relative contributions of each loss term.

## 4 Experiments

**Datasets and task descriptions.** We evaluate our approach on the 3DSSG dataset [45], a semantically enriched extension of 3RScan [44] designed for 3D semantic scene graph prediction *. This dataset consists of 1,553 real-world indoor scenes, annotated with 160 object categories and 26 predicate types, covering a wide range of household environments. We follow the standard train/validation split defined in the original benchmark. For details of the evaluation metrics, please refer to Section D of the Appendix.

### 4.1 Comparison with Other Works

We compare our method with the state-of-the-art approach VL-SAT [48], as well as several existing baselines, including SGPN [45] and SGFN [52], using their publicly available implementations or reported results. Our evaluation includes both quantitative performance metrics and qualitative

---

*We used the 3DSSG and 3RScan dataset [45] under the permission of its creators and authors. We contacted the authors by email and google form.

analysis, including t-SNE visualizations of object feature spaces and 3D scene graph visualizations. Additional experimental results are provided in Section D of the Appendix.

**Quantitative comparison.** Tables 2 and 3 show that our model achieves a new state-of-the-art performance on the 3DSSG benchmark across all evaluation metrics. The most significant improvements are observed in object classification. As reported in Table 2, our method improves object classification accuracy by 2–4% compared to VL-SAT [48]. This enhanced object representation contributes directly to better relationship classification, supporting our main assumption. Our method also consistently out-

| Model | Object | | Predicate | | Triplet | |
|---|---|---|---|---|---|---|
| | R@1 | R@5 | R@1 | R@3 | R@50 | R@100 |
| SGPN [45] | 49.46 | 73.99 | 86.92 | 94.76 | 85.38 | 88.59 |
| SGFN [52] | 53.36 | 76.88 | 89.00 | 97.71 | 88.59 | 91.14 |
| VL-SAT [48] | 55.93 | 78.06 | 89.81 | 98.46 | 89.35 | 92.20 |
| Ours | **59.53** | **81.20** | **91.27** | **98.48** | **91.40** | **93.80** |

Table 2: Quantitative results (%) on 3DSSG validation set. The **bold** denotes the best performance.

performs baselines in both SGCls and PredCls tasks, as reported in Table 3. Regardless of whether graph constraints are applied, we observe performance gains of 1–4% in both settings. These consistent improvements demonstrate not only the effectiveness of our approach over existing methods, but also provide empirical support for our assumption: enhancing object discrimination leads to more precise relationship prediction. The improvements in PredCls, where object labels are given and only relational reasoning is required, further substantiate this effect.

| Model | SGCls (w/ GC) | | | PredCls (w/ GC) | | | SGCls (w/o GC) | | | PredCls (w/o GC) | | |
|---|---|---|---|---|---|---|---|---|---|---|---|---|
| | R@20 | R@50 | R@100 | R@20 | R@50 | R@100 | R@20 | R@50 | R@100 | R@20 | R@50 | R@100 |
| SGPN [45] | 27.0 | 28.8 | 29.0 | 51.9 | 58.0 | 58.5 | 28.2 | 32.6 | 35.3 | 54.5 | 70.1 | 82.4 |
| Zhang et al. [63] | 28.5 | 30.0 | 30.1 | 59.3 | 65.0 | 65.3 | 29.8 | 34.3 | 37.0 | 62.2 | 78.4 | 88.3 |
| SGFN [52] | 29.5 | 31.2 | 31.2 | 65.9 | 78.8 | 79.6 | 31.9 | 39.3 | 45.0 | 68.9 | 82.8 | 91.2 |
| VL-SAT [48] | 32.0 | 33.5 | 33.7 | 67.8 | 79.9 | 80.8 | 33.8 | 41.3 | 47.0 | 70.5 | 85.0 | 92.5 |
| Ours | **36.1** | **37.7** | **37.8** | **70.2** | **82.0** | **82.6** | **38.1** | **46.1** | **52.5** | **73.3** | **87.8** | **94.6** |

Table 3: Quantitative results (%) of the SGCls and PredCls tasks, with and without graph constraints.

**Qualitative comparison.** Fig. 5 presents qualitative comparisons of predicted scene graphs between our method and VL-SAT [48]. VL-SAT frequently misclassifies visually similar but semantically distinct objects such as *cabinet* and *chair*, or *stool* and *garbage bin*, which leads to erroneous relationship predictions. These misclassifications result in hallucinated relationships that do not exist in the ground truth—for instance, predicting ⟨ *desk*, *standing on*, *floor* ⟩ or a spurious relation between *shelf* and *cabinet*. While such relationships might appear plausible given the visual similarity, they ultimately stem from incorrect object recognition, which propagates errors to the relational reasoning module. In contrast, our method correctly identifies object categories, thereby facilitating accurate and consistent relationship prediction. These examples highlight the critical role of object classification in downstream relation inference and reinforce the importance of robust object-level representations for reliable 3D scene graph construction.

To further investigate the underlying feature representations, we visualize the learned object embedding space using t-SNE for the ten most frequent object categories in the dataset (Fig. 6). Compared to VL-SAT, our approach yields more compact and well-separated clusters, particularly for structurally similar object pairs such as *ceiling–floor*, *wall–door*, and *curtain–window*. These results suggest that our object encoder learns more discriminative features, which provide a semantically stronger foundation for subsequent

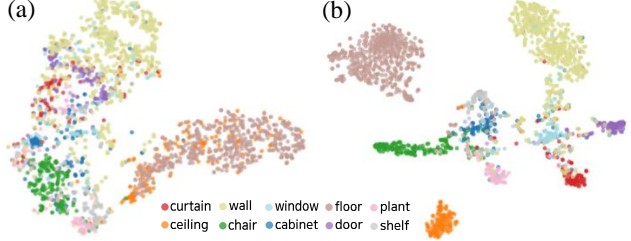

Figure 6: The t-SNE visualization of the object latent space of the VL-SAT [48] (a) and our model (b).

relationship classification. Overall, the qualitative evidence supports our claim that improving object discrimination leads to more accurate relational reasoning.

## 4.2 Discussions

| Model | OFL (ours) | Object | | Predicate | | Triplet | | SGCls | | PredCls | |
|---|---|---|---|---|---|---|---|---|---|---|---|
| | | R@1 | R@5 | R@1 | R@3 | R@50 | R@100 | R@20 | R@50 | R@20 | R@50 |
| SGPN [45] | ✗ | 47.37 | 72.00 | 88.60 | 97.15 | 85.83 | 89.06 | 22.9 | 24.0 | 63.9 | 75.3 |
| | ✓ | 54.49 | 75.02 | 90.10 | 98.06 | 88.83 | 91.16 | 29.8 | 31.0 | 68.2 | 79.0 |
| | | +7.12% | +3.02% | +1.50% | +0.91% | +3.00% | +2.10% | +6.9% | +7.0% | +4.3% | +3.7% |
| SGFN [52] | ✗ | 56.18 | 78.04 | 89.61 | 98.01 | 89.50 | 92.05 | 31.5 | 33.0 | 67.7 | 79.2 |
| | ✓ | 58.75 | 79.70 | 89.63 | 98.24 | 89.99 | 92.41 | 35.0 | 36.3 | 70.7 | 80.9 |
| | | +2.57% | +1.66% | +0.02% | +0.23% | +0.49% | +0.36% | +3.5% | +3.3% | +3.0% | +1.7% |
| VL-SAT [48] | ✗ | 55.68 | 78.06 | 89.81 | 98.45 | 89.43 | 92.22 | 32.0 | 33.5 | 67.8 | 80.0 |
| | ✓ | 59.30 | 80.67 | 90.48 | 98.51 | 90.40 | 93.03 | 34.9 | 36.6 | 70.6 | 81.7 |
| | | +3.62% | +2.61% | +0.67% | +0.06% | +0.97% | +0.81% | +2.9% | +3.1% | +2.8% | +1.7% |

Table 4: **Ablation studies on OFL.** ✗ and ✓ indicate whether OFL is not applied or applied, respectively. SGCls and PredCls are evaluated with graph constraints, and gray percentages indicate relative performance improvements when OFL is applied.

| GSE | BEG | LSE | Object | | Predicate | | Triplet | | SGCls | | PredCls | |
|---|---|---|---|---|---|---|---|---|---|---|---|
| | | | R@1 | mR@1 | R@1 | mR@1 | R@50 | mR@50 | R@50 | mR@50 | R@50 | mR@50 |
| | | | 58.02 | 20.77 | 90.55 | 50.36 | 90.19 | 61.79 | 43.8 | 36.5 | 85.7 | 68.3 |
| ✓ | | | 59.28 | 21.10 | 90.69 | 50.80 | **91.51** | 62.59 | 46.0 | 39.9 | 87.0 | 68.5 |
| ✓ | ✓ | | 59.49 | 22.17 | 90.65 | 53.81 | 91.18 | 64.83 | 45.7 | 43.0 | 86.7 | 73.2 |
| ✓ | ✓ | ✓ | **59.53** | **22.56** | **91.27** | **56.32** | 91.40 | **65.31** | **46.1** | **44.5** | **87.7** | **74.7** |

Table 5: **Ablation studies on proposed methods.** ✓ indicates the use of each component. All metrics include mean recall (mR), and both SGCls and PredCls are evaluated without graph constraints.

We conduct comprehensive ablation experiments to quantify the contribution of each proposed component to our model's performance, as summarized in Table 4 and 5.

**Plug-in our object encoder to other models.** To verify that our performance gains considerably stem from the object encoder, we integrated our pretrained encoder into three representative baselines: SGPN [45], SGFN [52], and VL-SAT [48], while keeping all other components unchanged. The results in Table 4 show consistent improvements across all frameworks. For SGPN [45], object recall increases at most 7% and other metrics also increased significantly. Most notably, even the state-of-the-art VL-SAT [48] benefits from our object encoder, where overall metrics are improved. These consistent enhancements across different architectures confirm that our pretraining strategy yields more discriminative and transferable object embeddings that directly benefit relationship prediction.

**Effectiveness of GSE.** Ablating the GSE component reveals its importance for object discrimination. Table 5 shows that removing GSE leads to noticeable degradation in overall recall metrics, with a drop of approximately 1–2%. This suggests that incorporating spatial proximity primarily benefits object representation quality and predicate reasoning, thereby supporting our assumptions.

**Effectiveness of BEG.** The ablation results demonstrate that BEG is crucial for accurate relationship modeling. When BEG is removed, performance drops significantly in overall metrics. Notably, all mR metrics decrease by approximately 1–4%, confirming that explicitly modeling directional asymmetry via controlled information flow between forward and reverse edges substantially improves contextual understanding, highlighting its importance in capturing the inherent directionality of semantic relationships between objects.

**Effectiveness of LSE.** As shown in Table 5, removing the LSE results in performance drops across all metrics. Especially, performance decreases by approximately 1-2% in SGCls and PredCls when this component is removed. This confirms that explicitly preserving information of spatial relation-

ship helps balance the representation learning between object semantics and spatial relationships, preventing the model from over-relying on object features alone.

The full model incorporating all proposed methods achieves the best performance across nearly all metrics, demonstrating their complementary nature in addressing the fundamental challenges of 3D scene graph prediction.

## 5 Conclusion

We propose an effective framework that leverages object features to enhance relationship prediction in 3D semantic scene graph prediction. Specifically, we introduce a simple yet effective pretraining strategy for learning a more discriminative object feature space, along with a novel Bidirectional Edge Gating mechanism designed to fully exploit this representation. In addition, by integrating auxiliary tasks and a dedicated relationship encoder, our approach outperforms the existing methods under the closed-vocabulary 3DSSG setting. Extensive experiments demonstrate the effectiveness of our approach, particularly highlighting significant improvements in the PredCls task. We believe this work establishes a solid foundation for future research in 3DSSG, highlighting the potential of integrating advances in object detection and classification to further advance 3D scene understanding.

## Acknowledgments

This work was supported by the National Research Foundation of Korea (NRF) grant funded by the Korea government (MSIT)(RS-2024-00456589) and Institute of Information & communications Technology Planning & Evaluation (IITP) grant funded by the Korea government (MSIT) (No. RS-2025-02263277 and RS-2022-00155911, Artificial Intelligence Convergence Innovation Human Resources Development (Kyung Hee University)).

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

# Appendix

**Overview.** The Appendix is structured as:

# A    Related Works

**Supervised Contrastive Learning.** Contrastive learning has recently emerged as a powerful paradigm for self-supervised representation learning. Gutmann *et al.* [19] laid the theoretical foundation by casting representation learning as density-ratio estimation between target and noise distributions. SimCLR [6] established the practical effectiveness of this paradigm through the NT-Xent objective, while MoCo [20] extended this idea by introducing a momentum encoder and a memory bank to effectively enlarge the set of negatives. Yeh *et al.* [56] subsequently introduced Decoupled Contrastive Learning (DCL), a variant designed to reduce negative–positive coupling by removing the positive term from the denominator. Contrastive objectives have also been successfully adapted to fully supervised settings: SupCon [23] exploits class labels directly; TSC [28] utilizes deterministic class centres to address long-tailed distributions; and ParCon [7] incorporates parametric centres within the MoCo framework. Motivated by the principles underlying DCL [56] and SupCon [23], we introduce a cross-modal supervised contrastive objective that leverages textual labels to learn a more discriminative object feature space, thereby enhancing predicate prediction accuracy.

**3D Point Cloud Understanding.** Early research on 3D point-cloud analysis sought affine-invariant representations directly from raw points to support downstream tasks. PointNet [35] introduced a shared multilayer perceptron with max-pool aggregation for global object descriptors, and Point-Net++ [36] refined this design through hierarchical farthest-point sampling and local neighbourhood grouping. Due to their computational efficiency, these models remain foundational for various 3D applications, and many 3DSSG pipelines still build upon them. Subsequent work has demonstrated that unsupervised objectives can yield even more transferable point descriptors. PointContrast [53] aligns local fragments from multiple scans of the same scene, while Hou *et al.* [21] exploit scene-level context by applying contrastive objectives on scene graphs, thereby improving data efficiency on indoor datasets. CrossPoint [1] extends such alignment to the cross-modal setting of point clouds and RGB images, and CLIP$^2$ [60] jointly aligns language, images, and point clouds. The latter, however, employs a fixed prompt vocabulary and does not explicitly address instance-level discrimination within a 3D scene. In this work, we address this limitation by proposing a supervised contrastive formulation that explicitly leverages available image and text annotations. Drawing inspiration from CLIP$^2$ [60], our object encoder fuses visual and spatial cues to produce more discriminative and semantically meaningful instance-level representations.

**3D Scene Graph Prediction.** Several approaches have attempted to incorporate relational or global context into 3D scene understanding. For instance, the pioneering SGPN [45] combined PointNet [35] with a graph neural network and introduced the widely adopted 3DSSG benchmark. Subsequent graph-based methods propagate features along adjacency edges to refine node embeddings and encode spatial priors, but their effectiveness remains constrained by the quality and discriminative power of initial object descriptors. Within 3D scene-graph generation, SGFN [52] introduced a feature-wise attention mechanism within a GNN framework, substantially outperforming geometry-only baselines relying solely on PointNet embeddings. Similarly, SGGpoint [61] leveraged edge-oriented graph convolutions to incorporate multi-dimensional geometric features explicitly into relationship modeling, while Zhang *et al.* [63] employed graph auto-encoders to explicitly integrate prior knowledge of object and predicate classes. More recent studies have explored alternative learning paradigms to address limitations inherent to fully supervised training. VL-SAT [48] leverages visual–linguistic auxiliary tasks during training, achieving state-of-the-art performance by enriching object and predicate representations. Koch *et al.* [26] employ a self-supervised reconstruction

objective as a regularizer to enhance the quality of learned latent representations, and 3D-VLAP [47] adopts a weakly supervised approach, aligning 3D data with 2D images and textual labels.

**Object-Centric Representation Learning.** Object-centric approaches have proliferated in both vision and NLP domains. In 2D scene-graph generation, KERN [5] leverages statistical object co-occurrence patterns to enhance relationship prediction accuracy while regularizing semantic predictions to address uneven distribution issues. Similar to the present framework, KERN relies on high-quality object detections from Faster R-CNN. For visual-relationship detection, ViP [29] implements a phrase-guided message-passing structure that jointly infers subject, predicate, and object elements. In knowledge-graph completion research, TransR [31] maps entities and relations into distinct embedding spaces, extending previous models such as TransE and TransH. The approach proposed in this work builds upon these foundational ideas while placing greater emphasis on learning highly discriminative object features, which subsequently serve as strong priors for predicate estimation.

**Open-Vocabulary Settings.** Recently, research utilizing VLM models such as OpenSeg [16] and InstructionBLIP [8] for Scene Graph Generation has been actively conducted. Typically, hybrid approaches that utilize both VLM/LLM and GNN have emerged. Open3DSG [27] is a representative example that performs knowledge distillation to lighter PointNet and GCN using well-trained VLMs[16, 8]. Specifically, they extract object features from scene images through OpenSeg and extract relationship features between two objects in images through InstructionBLIP. They perform knowledge distillation so that this information can be reflected in the node/edge features of GNN and the object/relationship PointNet respectively, pretraining the final scene graph model. During inference, they use CLIP for nodes and LLM for estimating relationships between nodes in an open-vocabulary setting. Meanwhile, Gu *et al.* [18] presented an approach that constructs Scene Graphs using only a combination of VLM/LLM. They primarily utilized LLaVA-7B, an LVLM, to detect objects from RGBD image sequences captured in scenes and set these as nodes in the scene graph. After node detection, they construct an MST (Minimum Spanning Tree) using the IOU of bounding boxes as weights for all 3D object pairs, and finally use LLMs such as GPT-4 to infer edges. By utilizing LVLM/LLM in both node and edge prediction processes, their work became pioneering research in scene graph generation in open-vocabulary settings, demonstrating the potential for expansion to various downstream tasks. Additionally, in 2025, Zhang *et al.* [62] extended the existing 3D Scene Graph Generation task to propose a VLM/LLM-based model that outperforms existing models, along with a dataset that can construct more practical scene graphs considering interactive objects and functional relationships. Similar other studies [39, 22, 59, 10, 49] are continuing research in directions similar to ConceptGraph [18], showing great potential for development in this field.

Our method departs from prior work in three key aspects. First, whereas most existing approaches primarily focus on predicate prediction, we prioritize learning highly discriminative and semantically robust object-level representations, which subsequently improve predicate inference. Second, in contrast to earlier contrastive pipelines [1, 53, 60], our encoder is explicitly optimized at the instance level, effectively capturing relational semantics critical to the demands of 3DSSG. Lastly, compared to those researches using only VLM/LLM to build scene graph, our study focused on closed-vocabulary settings to effectively verify our insights noted in Section 2, Observation of manuscript. Considering the fundamental differences between GNN-based and VLM/LLM-based methods, we chose a GNN-based approach to clearly and effectively verify our hypothesis: predicate classification errors are strongly correlated with incorrect or uncertain object label predictions. On the other hand, we excluded VLM/LLM from our choices to remove dependency on foundation model performance which will limit our ability to improve object encoder discrimination power.

# B  Details of Observations and Methodology

## B.1  Object Classification Entropy and Predicate Errors

We provide additional details on the empirical observation presented in the manuscript.

The conditional entropy $H(o \mid \mathbf{z})$ is computed based on the object classification distribution predicted by the model. Specifically, let $P(\hat{o}_i \mid \mathbf{z}_i)$ denote the class prediction distribution output by the GNN

for object $i$; the model's prediction is expressed as:

$$P(\hat{o}_i = c|\mathbf{z}_i^t) = \frac{\exp(f_g(\mathbf{z}_i^t)[c])}{\sum_{c'} \exp(f_g(\mathbf{z}_i^t)[c'])} \qquad (9)$$

where $\mathbf{z}_i^t$ is the object feature vector, and $f_g$ denotes the abstract representation of our object predictor. We then compute the entropy of this distribution, representing the uncertainty of our prediction, as follows:

$$H(\hat{o}_i|\mathbf{z}_i^t) = -\sum_c P(\hat{o}_i = c|\mathbf{z}_i^t) \log P(\hat{o}_i = c|\mathbf{z}_i^t) \qquad (10)$$

We examined the correlation between the entropy of predicted objects (subject and object) and predicate prediction accuracy. Specifically, for each predicted relation $\langle s, p, o \rangle$, we performed the following procedure: **(i)** We retain only instances where the subject ($s$) and object ($o$) are *both* classified correctly at Top-1, thus isolating the influence of object prediction confidence; **(ii)** We compute the *accumulated entropy* as:

$$E_{\mathrm{obj}} = \tfrac{1}{2}\big[H(s|\mathbf{z}_s^t) + H(o|\mathbf{z}_o^t)\big], \qquad (11)$$

which is the average conditional entropy of the two predicted objects (subject and object); **(iii)** We assign a binary predicate-error indicator $e$ as:

$$e = \begin{cases} 1, & \text{if } \hat{p} \neq p^\star, \\ 0, & \text{otherwise,} \end{cases} \qquad (12)$$

where $\hat{p}$ and $p^\star$ denote the predicted and ground-truth predicates, respectively. From the collected pairs $(E_{\mathrm{obj}}, e)$, we construct histograms of $E_{\mathrm{obj}}$ and compute corresponding predicate-error rates.

The predicate-error ratio for each bin is computed as $\sum e / \#\mathrm{pairs}$. The empirical finding that the predicate-error rate increases monotonically with object-classification entropy lends persuasive support to our central hypothesis: higher confidence in object predictions leads to more accurate predicate inference.

## B.2 Theoretical details of object feature learning.

Unlike existing pre-training approaches for 3D point clouds, this work employs supervised contrastive learning in which the positive term is deliberately omitted from the loss denominator. As noted in Appendix § A, prior work on multimodal contrastive learning for 3D objects typically relies on InfoNCE-style losses, similar to those used in SimCLR. The conventional motivation is to maximize mutual information $I(o; \mathbf{z})$ by minimizing the InfoNCE loss, which can be interpreted as the conditional entropy $H(o|\mathbf{z})$ with positive terms included in the denominator. However, as in DCL [56], the positive term in the denominator is deliberately omitted to enhance representational capacity within the self-supervised learning framework. This section provides theoretical evaluation; the precise loss definition is provided in the manuscript. To analyze the impact of the positive term in the denominator, consider a text-modal loss that contains this term, similar to previous works [23, 1, 60, 34, 7]. $\mathcal{L}_{\mathrm{text}}^i$ can be formulated as:

$$\mathcal{L}_i^{\mathrm{text}} = -s(\mathbf{z}_i^t, \mathbf{z}_{\mathrm{text}}^i)/\tau + \log U_i, \quad U_i = \sum_{a \in \mathcal{B}} \exp(s(\mathbf{z}_i^t, \mathbf{z}_{\mathrm{text}}^a)/\tau) \qquad (13)$$

$U_i$ can be decomposed into positive and negative sample terms, $\tilde{P}_i$ and $\tilde{N}_i$, respectively:

$$\tilde{N}_i = \sum_{n \in \mathcal{N}(i)} \exp(s(\mathbf{z}_i^t, \mathbf{z}_{\mathrm{text}}^n)/\tau), \quad \tilde{P}_i = U_i - \tilde{N}_i \qquad (14)$$

The following proposition 1 is presented. For clarity, all embedding vectors are assumed to be normalized.

**Proposition 1.** *There exists a multiplier $q_{text}^i$ in the gradient $\mathcal{L}_{text}^i$ analogous to the negative-positive coupling (NPC) term proposed in [56].*

$$-\nabla_{\mathbf{z}_i^t}\mathcal{L}_i^{text} = \frac{q_{text}^i}{\tau}\left(\mathbf{z}_{text}^i - \frac{1}{\tilde{N}_i}\sum_{n\in\mathcal{N}(i)}\exp(s(\mathbf{z}_i^t, \mathbf{z}_{text}^n)/\tau)\cdot\mathbf{z}_{text}^n\right) \tag{15}$$

where the multiplier $q_{\text{text}}^i$ is defined as:

$$q_{\text{text}}^i = \frac{\tilde{N}_i}{U_i} = \frac{\tilde{N}_i}{\tilde{P}_i + \tilde{N}_i} \tag{16}$$

*Proof.*

$$
\begin{aligned}
-\nabla_{\mathbf{z}_i^t}\mathcal{L}_i^{\text{text}} &= \frac{1}{\tau}\left(\mathbf{z}_{\text{text}}^i - \frac{1}{U_i}\sum_{a\in\mathcal{B}}\exp(s(\mathbf{z}_i^t, \mathbf{z}_{\text{text}}^a)/\tau))\cdot\mathbf{z}_{\text{text}}^a\right)\\
&= \frac{1}{\tau}\left(\mathbf{z}_{\text{text}}^i - \frac{1}{U_i}\sum_{p\in\mathcal{P}(i)}\exp(s(\mathbf{z}_i^t, \mathbf{z}_{\text{text}}^p)/\tau))\cdot\mathbf{z}_{\text{text}}^p - \frac{1}{U_i}\sum_{n\in\mathcal{N}(i)}\exp(s(\mathbf{z}_i^t, \mathbf{z}_{\text{text}}^n)/\tau))\cdot\mathbf{z}_{\text{text}}^n\right)\\
&= \frac{1}{\tau}\left(\mathbf{z}_{\text{text}}^i\left(1 - \frac{1}{U_i}\sum_{p\in\mathcal{P}(i)}\exp(s(\mathbf{z}_i^t, \mathbf{z}_{\text{text}}^p)/\tau))\right) - \frac{\tilde{N}_i}{U_i}\cdot\frac{1}{\tilde{N}_i}\sum_{n\in\mathcal{N}(i)}\exp(s(\mathbf{z}_i^t, \mathbf{z}_{\text{text}}^n)/\tau))\cdot\mathbf{z}_{\text{text}}^n\right)\\
&= \frac{1}{\tau}\frac{\tilde{N}_i}{U_i}\left(\mathbf{z}_{\text{text}}^i - \frac{1}{\tilde{N}_i}\sum_{n\in\mathcal{N}(i)}\exp(s(\mathbf{z}_i^t, \mathbf{z}_{\text{text}}^n)/\tau)\cdot\mathbf{z}_{\text{text}}^n\right)\\
&= \frac{q_{\text{text}}^i}{\tau}\left(\mathbf{z}_{\text{text}}^i - \frac{1}{\tilde{N}_i}\sum_{n\in\mathcal{N}(i)}\exp(s(\mathbf{z}_i^t, \mathbf{z}_{\text{text}}^n)/\tau)\cdot\mathbf{z}_{\text{text}}^n\right)
\end{aligned}
$$

Since $\mathbf{z}_{\text{text}}^p$ is identical for all $p\in\mathcal{P}(i)$, these terms can be consolidated as $\mathbf{z}_{\text{text}}^i$. □

Proposition 1 demonstrates that retaining the positive term in the denominator reproduces the critical issues identified in [56]: it simplifies the pretext task and generates disproportionately large gradients for instances with close positive and distant negative pairs. In the 3DSSG context, eliminating $q_{\text{text}}^i$ provides compelling analytical evidence supporting this hypothesis. Since every $\mathbf{z}_{\text{text}}^p$ with the same class label is identical, $q_{\text{text}}^i$ can be reformulated as:

$$q_{\text{text}}^i = \frac{\tilde{N}_i}{\tilde{P}_i + \tilde{N}_i} = \frac{\tilde{N}_i}{|\mathcal{P}(i)|\exp(s(\mathbf{z}_i^t, \mathbf{z}_{\text{text}}^i)/\tau) + \tilde{N}_i} \tag{17}$$

where $\mathcal{P}(i)$ represents the set of positive sample indices as defined in the manuscript.

Intuitively, insufficient attractive force between an anchor and its positives degrades class separation. This effect is particularly pronounced in 3DSSG, which contains challenging hard negatives such as distinguishing between *cabinet* and *kitchen cabinet*. Since their embeddings are inherently similar, their proximity to the anchor makes them exceptionally difficult to differentiate. Specifically, when $q_{\text{text}}^i \ll 1$ due to a large $\exp(s(\mathbf{z}_i^t, \mathbf{z}_{\text{text}}^i))$, the model fails both to effectively repel hard negatives $\mathbf{z}_{\text{text}}^h$ and to maintain adequate attraction to $\mathbf{z}_{\text{text}}^i$. This problematic behavior persists even in loss formulations that ignore class frequency.

Furthermore, $q_{\text{text}}^i$ is disproportionately suppressed for frequent classes, thereby weakening discrimination precisely where object accuracy is most critical to the predicate-error analysis. While retaining the positive term could theoretically help address long-tailed class imbalance in 3DSSG by balancing the impact of tail classes, it would obscure the fundamental relationship under investigation. Therefore, the positive term is deliberately omitted from the denominator. The same modification applies to $\mathcal{L}_{\text{visual}}^i$, further reinforcing the coherence of the loss design.

This analysis extends to variants where the class-frequency weight (i.e., $|\mathcal{P}(i)|$) is omitted, yielding:

$$\mathcal{L}^i_{\text{CE-like}} = -s(\mathbf{z}^t_i, \mathbf{z}^{c_i}_{\text{text}}) / \tau + \log U^c_i, \quad U^c_i = \exp(s(\mathbf{z}^t_i, \mathbf{z}^{c_i}_{\text{text}}) / \tau) + \sum_{c \neq c_i} \exp(s(\mathbf{z}^t_i, \mathbf{z}^c_{\text{text}}) / \tau) \quad (18)$$

which is algebraically equivalent to the conventional Cross Entropy loss.

Let $c$ denote the class label index and $c_i$ the class label index of the anchor. Under this definition, $\mathbf{z}^{c_i}_{\text{text}}$ serves as the unique positive for the anchor $\mathbf{z}^t_i$. The Cross Entropy-style loss is therefore given by Eq. 18, and the corresponding multiplier $q^i_{\text{CE-like}}$, following DCL [56], is formulated as:

$$q^i_{\text{CE-like}} = \frac{\sum_{c \neq c_i} \exp(s(\mathbf{z}^t_i, \mathbf{z}^c_{\text{text}}) / \tau)}{\exp(s(\mathbf{z}^t_i, \mathbf{z}^{c_i}_{\text{text}}) / \tau) + \sum_{c \neq c_i} \exp(s(\mathbf{z}^t_i, \mathbf{z}^c_{\text{text}}) / \tau)} \quad (19)$$

This formulation clearly exhibits the problems described in DCL [56]. It neither addresses the long-tailed distribution issues highlighted in Eq. 13 nor produces a sufficiently discriminative feature space, making it unsuitable for validating the hypotheses.

In Appendix § D.2, empirical comparisons between these conventional approaches and the proposed loss formulation are presented. The results conclusively demonstrate that excluding the positive term from the denominator is indeed an appropriate design choice.

## C  Implementation Details

**Object encoder pre-training.**  Object encoder pre-training is performed in PyTorch 1.12 (CUDA 11.3) on a single NVIDIA GeForce RTX 3060 Ti GPU. The network is optimized for 100 epochs using Adam, with a global batch size of 512, an initial learning rate of 0.01, and a cosine decay schedule to zero. Training takes roughly five hours to converge, and we retain the checkpoint achieving the highest cumulative validation accuracy summed across top-$K$ accuracies for $K \in \{1, 5, 10\}$ where classifier is CLIP[37]-based maximum similarity selection. For data preparation, we extract individual object point clouds from the 3DSSG scenes, translate them to the origin, randomly rotate them about the $z$-axis, and uniformly down-sample to 256 points. RGB-D images are cropped according to the VL-SAT pipeline [48]. Using CLIP [37], we select the four most similar images within the same scene that can be re-projected onto the object point cloud. As suitable images are not always available, the visual contrastive loss accommodates multiple images per positive sample. The original 3DSSG train/validation split is preserved throughout pre-training. The object encoder converges after approximately four hours under this setting. To evaluate classification accuracy in isolation, we train a three-layer MLP using Adam with a batch size of 256, a fixed learning rate of $1 \times 10^{-4}$, and standard Cross Entropy loss, without additional learning-rate scheduling.

**Scene graph prediction.**  All 3D Semantic Scene Graph experiments are conducted in PyTorch 1.12 with CUDA 11.3 on a single NVIDIA GeForce RTX 3090 GPU. The model is trained for 100 epochs using AdamW, with a global batch size of eight scenes and an initial learning rate of $1 \times 10^{-4}$ that decays following a cosine schedule. Each object point cloud is randomly down-sampled to 256 points. Both the proposed Global Spatial Enhancement (GSE) block and the Feature-wise Attention (FAT) relation head employ eight attention heads, and the GNN stack is unrolled for two iterations. The total loss is a weighted sum of the object ($\lambda_{obj}$), relation ($\lambda_{pred}$), and LSE ($\lambda_{lse}$) terms, with coefficients set to 0.1, 3.0, and 1.0, respectively. Training converges in approximately 36 hours, and the checkpoint with the highest validation *mean recall* (mR) at top-50 triplet prediction is selected for all quantitative and qualitative evaluations.

## D  Additional Experiments

### D.1  Metrics

Consistent with the 3DSSG protocol [45], we report top-$K$ *recall* (R@$K$) for both object and predicate recognition. Triplet scores are computed as the product of the subject, predicate, and object confidences; a triplet is deemed correct only when all three labels match the ground truth, and R@$K$ is then evaluated over these scores. We also adopt the two standard tasks introduced in [54]: Scene Graph Classification (SGCls), which evaluates complete triplets given ground-truth object boxes, and

| Model | Accuracy (%) | | | Mean Accuracy (%) | | |
|---|---|---|---|---|---|---|
| | Top-1 | Top-5 | Top-10 | Top-1 | Top-5 | Top-10 |
| Ours | **58.37** | **75.61** | 82.09 | 19.43 | 43.83 | 54.14 |
| Loss Eq.(13) | 52.99 | 73.19 | 81.36 | **21.77** | **45.48** | **56.68** |
| Loss Eq.(18) | 54.16 | 74.58 | **82.59** | 17.63 | 41.47 | 54.42 |

Table 6: Object classification performance of our method compared to loss variants of Eq.(13) and Eq.(18). Eq.(13) refers to our model with the positive term retained in the denominator, while Eq.(18) corresponds to the Cross Entropy-style loss. "Top-$k$ Accuracy" indicates overall classification accuracy, whereas "Top-$k$ Mean Accuracy" denotes class-balanced accuracy averaged over all 160 categories.

Predicate Classification (PredCls), which assesses predicate predictions under ground-truth object categories and boxes. For both tasks, we follow the evaluation protocol of Zhang *et al.* [63] and report *recall* at top-$K$ (R@$K$), counting a prediction as correct if the predicted subject, predicate, and object jointly match at least one ground-truth relation. Collectively, these metrics provide a comprehensive assessment of both object recognition accuracy and relational reasoning performance.

## D.2 Quantitative Results

| Model | Accuracy (%) | | | Mean Accuracy (%) | | |
|---|---|---|---|---|---|---|
| | Top-1 | Top-5 | Top-10 | Top-1 | Top-5 | Top-10 |
| VL-SAT (baseline) | 40.07 | 64.37 | 74.36 | 4.23 | 16.25 | 26.09 |
| Ours (full) | **58.37** | **75.61** | **82.09** | **19.43** | **43.83** | **54.14** |
| w/o visual modality | 56.72 | 72.74 | 79.21 | 16.66 | 37.45 | 47.14 |
| w/o affine regular. | 54.30 | 71.23 | 78.77 | 14.82 | 37.13 | 46.42 |
| w/o text modality | 16.10 | 39.34 | 53.74 | 1.05 | 3.68 | 7.36 |

Table 7: Object classification performance result of Ours and without each proposed methods. "Top-$k$ Accuracy" reports the overall object accuracy, whereas "Top-$k$ Mean Accuracy" averages accuracy across all 160 classes.

| GSE | BEG | LSE | Object | | Predicate | | Triplet | | SGCls | | PredCls | |
|---|---|---|---|---|---|---|---|---|---|---|---|---|
| | | | R@1 | mR@1 | R@1 | mR@1 | R@50 | mR@50 | R@50 | mR@50 | R@50 | mR@50 |
| | | | 58.02 | 20.77 | 90.55 | 50.36 | 90.19 | 61.79 | 43.8 | 36.5 | 85.7 | 68.3 |
| ✓ | | | 59.28 | 21.10 | 90.69 | 50.80 | **91.51** | 62.59 | 46.0 | 39.9 | 87.0 | 68.5 |
| | ✓ | | 58.67 | 21.50 | 91.12 | 50.39 | 90.63 | 64.00 | 44.2 | 38.9 | 86.6 | 69.5 |
| | | ✓ | 57.49 | 20.21 | 90.73 | 48.56 | 90.53 | 57.90 | 43.3 | 36.3 | 86.3 | 65.9 |
| ✓ | ✓ | | 59.49 | 22.17 | 90.65 | 53.81 | 91.18 | 64.83 | 45.7 | 43.0 | 86.7 | 73.2 |
| ✓ | | ✓ | 59.47 | 21.42 | **91.50** | 50.48 | 91.24 | 64.66 | **46.5** | 39.9 | 87.3 | 71.8 |
| | ✓ | ✓ | 57.72 | 20.43 | 91.38 | 50.47 | 90.85 | 60.90 | 43.9 | 38.1 | 86.3 | 73.2 |
| ✓ | ✓ | ✓ | **59.53** | **22.56** | 91.27 | **56.32** | 91.40 | **65.31** | 46.1 | **44.5** | **87.7** | **74.7** |

Table 8: **Ablation studies on proposed methods.** ✓ indicates the use of each component. All metrics include mean recall (mR), and both SGCls and PredCls are evaluated without graph constraints.

**Impact of removing positive term in denominator.** As shown in Table 6, the proposed loss achieves the highest Top-1 accuracy when compared to both loss functions of Eq. 13 and Eq. 18. In contrast, mean accuracy (mA@$k$) is higher for the loss variant with the positive term retained, which aligns with the theoretical analysis presented in Appendix § B.2. These empirical results are consistent with the analytical justification for omitting the positive term, as this approach enhances object-level discriminability, which is fundamental to the central hypothesis. The loss formulation of Eq. 18 exhibits lower performance on both accuracy metrics, corresponding to the limitations identified in the theoretical analysis. The combined theoretical and empirical observations provide evidence supporting the effectiveness of the proposed loss formulation and the underlying assumption.

| Model | Predicate | | | | | | Triplet | | | |
| --- | --- | --- | --- | --- | --- | --- | --- | --- | --- | --- |
| | Head | | Body | | Tail | | Unseen | | Seen | |
| | mR@3 | mR@5 | mR@3 | mR@5 | mR@3 | mR@5 | R@50 | R@100 | R@50 | R@100 |
| SGPN [45] | **96.66** | 99.17 | 66.19 | 85.73 | 10.18 | 28.41 | 15.78 | 29.60 | 66.60 | 77.03 |
| SGFN [52] | 95.08 | **99.38** | 70.02 | 87.81 | 38.67 | 58.21 | 22.59 | 35.68 | 71.44 | 80.11 |
| VL-SAT [48] | 96.31 | 99.21 | 80.03 | 93.64 | 52.38 | 66.13 | 31.28 | 47.26 | 75.09 | 82.25 |
| Ours | 96.25 | 99.13 | **84.92** | **95.34** | **56.66** | **70.33** | **32.24** | **48.25** | **79.70** | **86.18** |

Table 9: Quantitative results (%) of Top-K mean Recall (mR). For each split (Head, Body, Tail) we report *Predicate mR@3* and *Predicate mR@5*. Triplet Recall is given for *Unseen* and *Seen* relations at two cut-offs.

| Model | SGCls | | | PredCls | | |
| --- | --- | --- | --- | --- | --- | --- |
| | mR@20 | mR@50 | mR@100 | mR@20 | mR@50 | mR@100 |
| KERN [5] | 9.5 | 11.5 | 11.9 | 18.8 | 25.6 | 26.5 |
| SGPN [45] | 19.7 | 22.6 | 23.1 | 32.1 | 38.4 | 38.9 |
| SGFN [52] | 20.5 | 23.1 | 23.1 | 46.1 | 54.8 | 55.1 |
| Zhang *et al.* [63] | 24.4 | 28.6 | 28.8 | 56.6 | 63.5 | 63.8 |
| VL-SAT [48] | 31.0 | 32.6 | 32.7 | 57.8 | 64.2 | 64.3 |
| **Ours** | **35.3** | **37.7** | **37.9** | **58.8** | **66.1** | **66.4** |

Table 10: Quantitative results (%) of Top-K mean Recall (mR) on SGCls and PredCls with graph constraints. The **bold** denotes the best performance.

| Model | Object | | Predicate | | Triplet | |
| --- | --- | --- | --- | --- | --- | --- |
| | mR@1 | mR@5 | mR@1 | mR@3 | mR@50 | mR@100 |
| SGPN [45] | 10.86 | 27.54 | 32.01 | 55.22 | 41.52 | 51.92 |
| SGFN [52] | 21.24 | 46.68 | 41.89 | 70.82 | 58.37 | 67.61 |
| VL-SAT [48] | 21.41 | 46.14 | 54.03 | **77.67** | 65.09 | 73.59 |
| Ours | **22.55** | **48.10** | **56.32** | 76.26 | **65.31** | **74.54** |

Table 11: Quantitative results (%) of Top-K mean Recall (mR). The **bold** denotes the best performance.

**Ablation studies on object encoder.** Table 7 shows that substituting VL-SAT's point encoder with our multimodal object encoder yields a substantial boost in classification performance: Top-1/5/10 accuracy improved by approximately 8–20%, and the corresponding mean-accuracy figures nearly doubled. The ablation rows confirm that each component of the encoder contributes positively. Removing the visual branch decreases Top-1 accuracy by 1.65%, while discarding the affine-regularization term causes a larger 4.0% drop, indicating that redundancy reduction stabilizes instance-level cues. Eliminating the text branch has the most dramatic effect: Top-1 accuracy drops significantly to 16.10% and mean accuracy to 1.05%, highlighting the importance of language supervision for disambiguating geometrically similar classes. These results corroborate our claim that jointly leveraging visual, textual, and geometric signals produces sharper object posteriors, which in turn drive the gains observed in downstream scene-graph metrics.

**Detailed ablation studies on proposed methods.** Table 8 presents a comprehensive ablation study examining the performance impact of different architectural components, excluding the object feature encoder. All possible component combinations were systematically evaluated to identify their individual and combined contributions.

The Local Spatial Enhancement (LSE) component demonstrates complex performance interactions across different configurations. When LSE is used in isolation, performance metrics generally

| Model | Object | | Predicate | | Triplet | | SGCls | | PredCls | |
|---|---|---|---|---|---|---|---|---|---|---|
| | R@1 | mR@1 | R@1 | mR@1 | R@50 | mR@50 | R@50 | mR@50 | R@50 | mR@50 |
| w/o gating | 59.34 | **23.43** | 91.13 | 55.41 | 91.20 | 64.40 | 45.4 | 43.2 | 87.5 | 73.3 |
| w/ gating (ours) | **59.53** | 22.56 | **91.27** | **56.32** | **91.40** | **65.31** | **46.1** | **44.5** | **87.7** | **74.7** |

Table 12: **Ablation studies on Bidirectional Gating.** "w/o gating" refers to the model without gating on reverse edges in the bidirectional message passing, while "w/ gating (ours)" is our final model with gating applied to reverse edges. SGCls and PredCls are evaluated without graph constraints.

| Inputs | | | Object | | Predicate | | Triplet | | SGCls | | PredCls | |
|---|---|---|---|---|---|---|---|---|---|---|---|---|
| geo. | obj. | sub. | R@1 | R@5 | R@1 | R@3 | R@50 | R@100 | R@20 | R@50 | R@20 | R@50 |
| ✓ | | | 59.09 | 80.35 | 90.53 | 98.45 | 90.91 | 93.38 | 34.6 | 36.2 | 69.9 | 81.3 |
| ✓ | ✓ | ✓ | **59.53** | **81.20** | **91.27** | **98.48** | **91.40** | **93.80** | **36.1** | **37.7** | **70.2** | **82.0** |

Table 13: **Ablation studies on input of relationship encoder.** Checkmarks indicate whether each feature (object, subject, geometry) is used or not. SGCls and PredCls are evaluated with graph constraints.

decrease except for PredCls and *recall* metrics for predicates and triplets. However, when combined with Global Spatial Enhancement (GSE), LSE contributes to substantial performance improvements compared to using GSE alone. Conversely, the combination of LSE with Bidirectional Edge Gating (BEG) results in performance degradation. These observations suggest that GSE and LSE operate synergistically to improve performance across most metrics. From an analytical perspective, this pattern indicates that excessive emphasis on local information without global contextual understanding may lead to error propagation throughout the prediction pipeline, with BEG potentially amplifying this effect when global context is absent.

In contrast, GSE exhibits consistent performance improvements across all component combinations. Particularly when paired with BEG, GSE facilitates significant enhancement in *mean recall* metrics, with improvements of 3–5%. This suggests that incorporating global geometric information of objects within a scene plays a critical role in overall model performance.

BEG demonstrates consistent performance gains of 2–4% in *mean recall* metrics across most configurations, with the exception of its isolated combination with LSE. These improvements become particularly pronounced when BEG is implemented alongside GSE. This effectiveness can be attributed to BEG's capacity to simultaneously process bidirectional edge information, thereby enhancing the model's ability to encode and utilize local scene relationships.

The empirical results demonstrate that the optimal configuration incorporates all three components—GSE, LSE, and BEG—resulting in superior performance across evaluation metrics. This finding underscores the importance not only of extracting high-quality object features but also of effectively leveraging these features through complementary architectural components for scene graph generation.

**Performance on additional metrics.** Table 9 presents the quantitative evaluation results, reporting mR@$K$ for Head, Body, and Tail predicates ($K = 3, 5$), along with R@$K$ for seen and unseen triplets ($K = 50, 100$). The proposed method demonstrates consistent performance improvements across the majority of metrics. Notably, mR@$K$ for Tail predicates—typically the most challenging category due to data sparsity—shows an improvement of more than 4% over the previous highest performing method. Additionally, R@$K$ on *unseen* triplets increases by approximately one percentage point. These improvements, achieved despite the pronounced long-tailed class distribution characteristic of the 3DSSG dataset [45], indicate enhanced generalization capabilities of the proposed approach compared to existing methods.

Table 10 presents additional comparative results against all publicly available baselines on SGCls and PredCls tasks, reporting mR@$K$ for $K = 50, 100$. The experimental results demonstrate consistent improvements, with performance gains of 1–5% across all evaluation settings. The most significant improvement is observed in the SGCls *mean recall* metric, where the proposed method achieves a 4–5% increase relative to the previous best performing approach. These consistent improvements

across multiple metrics provide empirical evidence for the effectiveness of the proposed object-centric approach and its beneficial impact on subsequent relational reasoning tasks.

Table 11 completes the analysis by reporting Top-$K$ *mean recall* for objects, predicates, and full triplets. Because these metrics average over all classes, they are a direct indicator of a model's ability to cope with the severe long-tailed label distribution of 3DSSG. Our method improves every score by a further 1–2 %—which represents significant progress on this benchmark. Except for predicate mR@3, the gains are uniform across object, predicate, and triplet levels, suggesting that the discriminative object features learned by our encoder translate into more reliable relational reasoning. Taken together with the results in Tables 9 and 10, these findings demonstrate the generalization capabilities of the proposed approach relative to existing methods.

**Impact of gating in BEG.** Table 12 shows that applying our gating mechanism to reverse edges slightly improves or maintains all R@$K$ metrics, while consistently increasing nearly all mR@$K$ scores by 0.8–1.5%. Bidirectional message passing without controlled modulation may oversaturate node representations with redundant signals; our gating mechanism addresses this by selectively attenuating reverse-edge messages, thereby preserving directionally relevant information. The consistent improvements observed in predicate, triplet, and scene-level mR@$K$ metrics confirm that this modulation particularly benefits long-tail relations, yielding more balanced performance without sacrificing overall accuracy.

**Impact of semantic cues in relationship encoding.** Unlike previous approaches that relied solely on geometric information for relationship modeling, our relationship encoder integrates rich semantic features from our object encoder alongside geometric metadata. Table 13 shows that incorporating object-level semantics leads to a consistent improvement across all evaluation metrics. In particular, Predicate R@1 improves from 90.53 to 91.27, highlighting that semantic cues from object identity help the model better resolve predicate ambiguities. This result supports our initial hypothesis that combining geometric and semantic information enables more accurate and context-aware relationship reasoning, especially in cases where object semantics strongly constrain the set of plausible predicates.

| Method | SGCls | | | | PredCls | | | |
|---|---|---|---|---|---|---|---|---|
| | R@20 | mR@20 | R@50 | mR@50 | R@20 | mR@20 | R@50 | mR@50 |
| 3D-VLAP [47] | 18.7 | 11.3 | 21.8 | 13.9 | 53.5 | 31.8 | 64.4 | 41.2 |
| Ours w/ weakly-sup. | 25.5 | 18.1 | 28.2 | 21.2 | 61.5 | 38.8 | 69.7 | 45.4 |
| **Ours** w/ fully-sup. | **36.1** | **35.3** | **37.7** | **37.7** | **70.2** | **58.8** | **82.0** | **66.1** |

Table 14: **Extensibility to weakly-supervised settings.** "w/ weakly-sup." denotes the weakly supervised variant of our method and "w/ fully-sup." denotes our original setting(fully-supervised manner). Under weak supervision, our approach outperforms 3D-VLAP [47] on both R@K and mR@K. However, it does not match our fully supervised model, likely due to the inherent noise in weakly supervised signals arising from imprecise triplet labels.

**Extensibility to weakly-supervised setting.** As mentioned in Section. A, weakly-supervised approach also seems feasible for 3D scene graph generation task. Following 3D-VLAP [47], we adopted weakly-supervised setting to our approach and check whether our hypothesis also works with other training scheme. In Table. 14, proposed method also performs well with weakly-supervised setting, overwhelming original 3D-VLAP with huge margin. This result provides us useful insights that our hypothesis also valid in other training schemes. Especially, the mR@K showed huge gap compared to 3D-VLAP, which shows promising aspects to address long-tailed distribution problem with reducing extensive human annotation labor. However, It could not exceeded fully-supervised settings suggested in manuscript. We assume that weakly-supervised siginals(pseudo-labels selected by CLIP) made errors, causing somewhat tricky to learn proper insights.

## D.3 Qualitative Results

**Cosine heatmap visualization of object features.** Figure 7 visualizes the class-level cosine similarity matrices of object embeddings learned by our encoder (a) and the VL-SAT baseline (b) for ten representative categories. In our matrix, diagonal elements are nearly saturated ($\geq 0.95$), while off-diagonal cells quickly drop below 0.35, producing a clear block-diagonal pattern that indicates

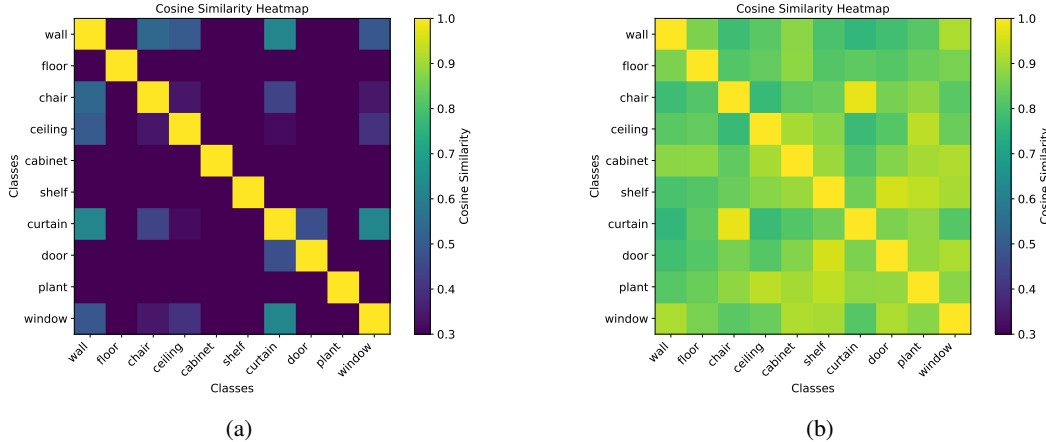

(a)              (b)

Figure 7: Qualitative analysis of object feature space. (a) Class-level cosine similarity of Ours, (b) is for same as that of VL-SAT [48].

high intra-class cohesion and strong inter-class separation. Particularly confusable pairs such as *cabinet–shelf* and *wall–ceiling* remain well below 0.40, suggesting the encoder captures fine-grained geometric cues. By contrast, the VL-SAT matrix is notably diffuse: most off-diagonal similarities range from 0.6–0.8, and visually similar classes (*cabinet*, *shelf*, *door*) share nearly identical colors with diagonal elements. The sharper contrast in our embedding space supports our hypothesis that a more discriminative object feature space directly translates into higher predicate and triplet accuracy.

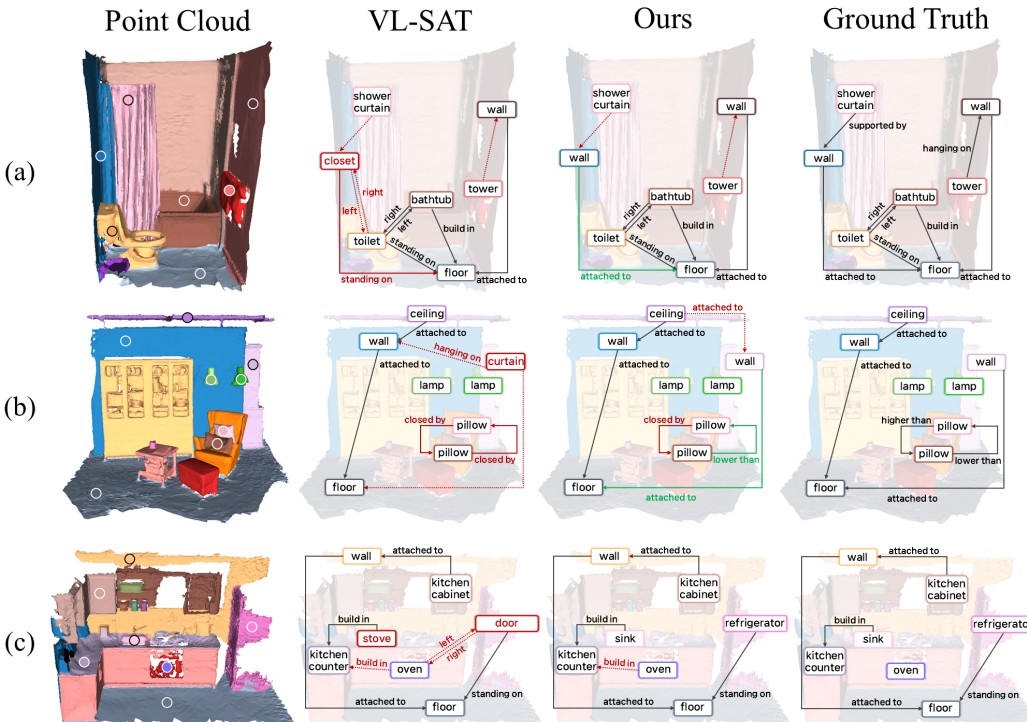

Figure 8: **3D scene graph visualizations.** → indicates true positive relations that are correctly predicted. → denotes false positives, where the model predicts an incorrect predicate for an existing relation. --→ represents either false negatives—missed ground-truth relations—or hallucinated relations that do not exist in the ground truth.

**Additional visualizations of 3D scene graph.** We provide additional predicted scene graphs from our proposed method in Fig. 8. As hypothesized, object misclassification significantly impacts predicate prediction accuracy within the scene graph. Scene (a) illustrates how information propagated through the robust object encoder effectively filters out non-existent relationships. VL-SAT incorrectly classifies a wall as a closet, consequently leading to erroneous relationship predictions involving the toilet and floor. This misclassification results in VL-SAT inferring relationships that are absent in the ground truth data, highlighting the importance of accurate object classification for downstream relationship prediction. In scene (b), VL-SAT exhibits a classification error by identifying a wall as a curtain. While these objects share similar geometric structures and only differ substantially in scale, accurate classification requires precise consideration of dimensional attributes. The robustness of the proposed object encoder enables clear distinction between these visually similar objects, resulting in reduced predicate error rates compared to VL-SAT. In scene (c), the proposed object encoder demonstrates notable robustness in classification performance. Objects that were misclassified by VL-SAT, specifically the refrigerator and sink, are correctly identified by the proposed approach. This accurate classification enables the model to more successfully reason about relationships between the refrigerator and oven compared to VL-SAT. Furthermore, the correct differentiation between stove and sink—objects with potentially confounding morphological similarities—provides additional evidence for the robustness of the proposed object encoder.

# E Limitations and Further Works

The present study strictly follows the evaluation protocol established in 3DSSG and consequently does not incorporate 3D object detection capabilities. As a result, the proposed approach cannot be directly applied to real-world settings where 3D object detection must be performed. Furthermore, the method requires the entire scene to be available for scene graph generation, as it does not support incremental graph updates—a limitation for practical deployment in real-world scenarios.

For future work, we aim to develop an integrated framework that combines 3D object detection with an incremental scene graph generation module. This study has demonstrated that object representations play a crucial role in relationship reasoning. This finding suggests that leveraging existing high-performance object representation methods could significantly enhance overall scene graph generation performance. The established connection between object representation quality and relationship prediction accuracy also provides valuable insights for improving model performance when integrated with 3D object detection systems. Our subsequent research will explore effective methods for combining these components to develop more practical off-the-shelf scene graph generation algorithms.

