# OpenReview forum: "Object-Centric Representation Learning for Enhanced 3D Semantic Scene Graph Prediction"
_NeurIPS.cc/2025/Conference — NeurIPS 2025 poster_

### Official Review · Reviewer_dCsV · 2025-06-12

**Clarity:** 2
**Significance:** 2
**Originality:** 2
**Rating:** 3
**Confidence:** 5

**Summary:**

This paper proposes a new method for scene graph generation. Given existing scene point clouds with instance masks, the method could infer the relationships of instances through graph neural networks. This paper has some improvements in object feature learning and relationship feature learning. The experiment results prove that this paper has improvements over the previous baseline methods and the ablation study proves the effectiveness of the proposed contributions.

**Questions:**

My questions are in the weaknesses part. I will appreciate any responses to my questions. I will raise my points to positive if the authors could address my concerns.

**Ethical Concerns:**

["NO or VERY MINOR ethics concerns only"]

**Final Justification:**

After rebuttal and discussion with the authors, I tend to keep my score. The paper only focuses on close-set scene graph generation, and lacks the analysis of VLM-based open-set scene graph generation in their original submitted paper.

**Limitations:**

yes

**Quality:**

2

**Strengths And Weaknesses:**

Strengths:
1. the motivations of this paper look persuasive.
2. the experiment results prove the improvements over previous baselines.

Weaknesses:
1. the main concern is that, for this scene graph generation task, after the emergence of VLM/LLM, the methods or approaches for this tasks are totally different and many new methods come out. I would suggest the authors keep track of the concurrent scene graph generation papers and give a detailed related work analysis in this field in the main paper.
2. following point 1, I notice that the baseline methods seem to be old methods. So if possible, it would be great if the authors could elaborate why for the scene generation tasks, the community still needs this series of GNN methods? is it better than those VLM based methods? or is it more robust? any reasons would be appreciated.
3. the paper method part is too complicated. I would suggest the author to simplify it and avoid excessive inline math formulas in this part. It hurts the readability.

---

> ### Author Rebuttal · Authors · 2025-07-29
>
> We appreciate reviewer's review and have made every effort to address the concerns reviewer have raised.
>
> # [Q1, W1] Detailed Related Works of the concurrent papers
> As reviewer mentioned, the emergence of recent VLM/LLM models has significantly transformed approaches in the Scene Graph Generation field, with many new studies being published.
>
> ### **Recent Research Trends Based on VLM/LLM**
>
> Recently, research utilizing VLM models such as OpenSeg [A] and InstructionBLIP [B] for Scene Graph Generation has been actively conducted.
>
> Typically, hybrid approaches that utilize both VLM/LLM and GNN have emerged. Open3DSG [C] is a representative example that performs knowledge distillation to lighter PointNet and GCN using well-trained VLMs (OpenSeg, InstructionBLIP). Specifically, they extract object features from scene images through OpenSeg and extract relationship features between two objects in images through InstructionBLIP. They perform knowledge distillation so that this information can be reflected in the node/edge features of GNN and the object/relationship PointNet respectively, pretraining the final scene graph model. During inference, they use CLIP for nodes and LLM for estimating relationships between nodes in an open-vocabulary setting.
>
> Meanwhile, Gu et al. [D] presented an approach that constructs Scene Graphs using only a combination of VLM/LLM. They primarily utilized LLaVA-7B, an LVLM, to detect objects from RGBD image sequences captured in scenes and set these as nodes in the scene graph. After node detection, they construct an MST (Minimum Spanning Tree) using the IOU of bounding boxes as weights for all 3D object pairs, and finally use LLMs such as GPT-4 to infer edges. By utilizing LVLM/LLM in both node and edge prediction processes, their work became pioneering research in scene graph generation in open-vocabulary settings, demonstrating the potential for expansion to various downstream tasks.
>
> Additionally, in 2025, Zhang et al. [E] extended the existing 3D Scene Graph Generation task to propose a VLM/LLM-based model that outperforms existing models, along with a dataset that can construct more practical scene graphs considering interactive objects and functional relationships. Similar other studies [F, G, H, I, J] are continuing research in directions similar to ConceptGraph [D], showing great potential for development in this field.
>
> Compared to those researches using only VLM/LLM to build scene graph, our study focused on closed-vocabulary settings to effectively verify our insights noted in **Section 2 Observation** (Detailed explanations are in response to **[Q2, W2]**).
>
> As reviewer's comment, we will add a section on VLM/LLM-based approaches in the Manuscript to more comprehensively cover the latest trends in the current scene graph generation field.
>
> [A] Ghiasi, Golnaz, et al. "Scaling open-vocabulary image segmentation with image-level labels." *European conference on computer vision*. Cham: Springer Nature Switzerland, 2022.
>
> [B] Dai, Wenliang, et al. "Instructblip: Towards general-purpose vision-language models with instruction tuning." *Advances in neural information processing systems* 36 (2023): 49250-49267
>
> [C] Koch, Sebastian, et al. "Open3dsg: Open-vocabulary 3d scene graphs from point clouds with queryable objects and open-set relationships." Proceedings of the IEEE/CVF Conference on Computer Vision and Pattern Recognition. 2024
>
> [D] Gu, Qiao, et al. "Conceptgraphs: Open-vocabulary 3d scene graphs for perception and planning." 2024 IEEE International Conference on Robotics and Automation (ICRA). IEEE, 2024
>
> [E] Zhang, Chenyangguang, et al. "Open-vocabulary functional 3d scene graphs for real-world indoor spaces." Proceedings of the Computer Vision and Pattern Recognition Conference. 2025.
>
> [F] Rotondi, Dennis, et al. "FunGraph: Functionality Aware 3D Scene Graphs for Language-Prompted Scene Interaction." *arXiv preprint arXiv:2503.07909* (2025).
>
> [G] Kassab, Christina, et al. "The bare necessities: Designing simple, effective open-vocabulary scene graphs." *arXiv preprint arXiv:2412.01539* (2024).
>
> [H] Zemskova, Tatiana, and Dmitry Yudin. "3dgraphllm: Combining semantic graphs and large language models for 3d scene understanding." *arXiv preprint arXiv:2412.18450* (2024).
>
> [I] Devarakonda, Venkata Naren, et al. "Orionnav: Online planning for robot autonomy with context-aware llm and open-vocabulary semantic scene graphs." *arXiv preprint arXiv:2410.06239* (2024).
>
> [J] Werby, Abdelrhman, et al. "Hierarchical open-vocabulary 3d scene graphs for language-grounded robot navigation." *First Workshop on Vision-Language Models for Navigation and Manipulation at ICRA 2024*. 2024.
>
> # [Q2, W2] The reason for choosing GNN over VLM/LLM-based Methods
>
> ### **Differences between GNN and VLM/LLM-based Methods**
> GNN-based methods and VLM/LLM-based methods each have distinct characteristics in scene graph generation. GNN-based approaches excel at learning structured node and edge features with relatively small parameter counts, making them efficient for **closed-vocabulary settings**. However, they may show limitations in tasks requiring high generalization power. In contrast, VLM/LLM-based methods leverage pre-trained foundation models to achieve strong generalization capabilities, particularly in **open-vocabulary settings**, but come with inherent limitations such as hallucination, prompt-dependent answer quality, and high computational complexity.
>
> ### **VLM/LLM as a Tool for Open-Vocabulary Extension**
> We believe that VLM/LLM serves not as a replacement for GNN, but rather as a means to extend to open-vocabulary settings. This perspective is well demonstrated by Open3DSG [C], which utilizes both GNN and VLM/LLM models. Rather than viewing these methods as mutually exclusive, they can be appropriately utilized together based on specific requirements.
>
> ### **Our Research Approach**
> Our research was specifically designed to verify the observation presented in **Section 2** in a **closed-vocabulary setting**. Given the fundamental differences between GNN and VLM/LLM-based methods mentioned above, we chose a GNN-based approach to clearly and effectively verify our hypothesis: predicate classification errors are strongly correlated with incorrect or uncertain object label predictions. Using VLM/LLM would have introduced dependency on foundation model performance, limiting our ability to improve object encoder discrimination power.
>
> While direct performance comparison with VLM-based methods [D, E] is challenging due to different datasets and tasks, Open3DSG, which combines GNN and VLM/LLM methods, provides Top-K Recall metrics for Object/Predicate/Triplet as shown in the table below. These results indicate that open-vocabulary setting research primarily focuses on generalization performance, which is distant from addressing our hypothesis. However, we are confident that the insights gained from our closed-vocabulary research will serve as important clues for achieving better performance in future open-vocabulary scene graph generation.
>
> |Model|Obj. R@5|Obj. R@10|Pred. R@3|Pred. R@5|Tri. R@50|Tri. R@100|
> |---|---|---|---|---|---|---|
> |Open3DSG|0.57|0.68|0.63|0.70|0.64|0.66|
> |Ours|**0.81**|**0.88**|**0.98**|**0.99**|**0.91**|**0.94**|
>
> # [Q3, W3] Excessive inline math formulas
> We agree that the excessive use of inline math formulas is hampering readability and would like to present specific revision plans to address this issue.
>
> ### **Main Revision Direction**
> Based on the reviewer's comment, we determined that the formulas in the BEG and GSE sections are most significantly affecting readability, and we will revise them as follows:
>
> ### **Section 3.3 GSE Revision (L-214 ~ L-219)**
> First, we will remove unnecessary redundant definitions. We confirmed that the object's center vector $\mathbf{c}\_{i}$ and the $\mathbf{\mu}\_{i}$ (previously defined in **Section 3.2** of the manuscript) are semantically identical. Therefore, we will unify the distance calculation as $d\_{ij} = || \boldsymbol{\mu}\_{i} - \boldsymbol{\mu}\_{j} ||\_2$ and remove the unnecessary $\mathbf{r}_{ij}$ notation.
>
> ### **Section 3.3 BEG Revision (L-227 ~ L-237)**
> We will remove and unify redundant notations such as $\bar{\mathbf{z}}\_i$ and $\mathbf{f}\_i^{node}$, $\tilde{\mathbf{z}}\_{ij}^e$ and $\mathbf{f}\_{ij}^{edge}$. Additionally, we will separate the inline formulas related to node and edge updates into separate equations to improve readability:
>
> $\bar{\mathbf{z}}\_i \leftarrow \mathrm{LN}(\mathrm{MLP}(\mathrm{CAT}(\bar{\mathbf{z}}\_i, \sigma(a\_i))))$
>
> $\mathbf{z}^e\_{ij} \leftarrow \mathrm{MLP}\left(\mathrm{CAT}\left(\bar{\mathbf{z}}\_i, \mathbf{z}^{e}\_{ij}, \tilde{\mathbf{z}}^{e}\_{ij}, \bar{\mathbf{z}}\_j\right)\right)$
>
> ### **Additional Improvements**
>
> We will organize the main text explanations more concisely to minimize the use of inline formulas. To enhance intuitive understanding of how GSE and LSE operate, we will include additional diagrams in the Appendix or modify **Figure 4** to improve visual comprehension.
>
> Through these revisions, we expect to significantly improve the readability of the method section while maintaining technical accuracy. We thank reviewer once again for reviewer's valuable feedback.

---

> > ### Comment · Reviewer_dCsV · 2025-08-01
> >
> > Thanks for the rebuttal from the authors.
> >
> > 1. I strongly recommend the authors consider adding llm/vlm based scene graph generation into related works.
> >
> > 2. In your table above, you do not mention the test dataset and the experiment details. Also some numbers are like 0.98 & 0.99. How should I interpret those results?
> >
> > I will increase my score if the authors could address my concerns.

---

> > > ### Author Response · Authors · 2025-08-01
> > >
> > > We appreciate the reviewer's careful attention to these details.
> > >
> > > 1.
> > > We appreciate this valuable suggestion. We will definitely add LLM/VLM-based scene graph generation to the Related Works section in the final publication.
> > >
> > > 2.
> > > We apologize for the insufficient explanation of details. The test dataset is the 3DSSG dataset, specifically using the validation split provided for evaluation. The experimental settings in the table from our **[Q2, W2]** response are identical to those mentioned in **Section 4 (Datasets and task descriptions)** of our Manuscript. To briefly explain:
> > > - Evaluation was conducted on the 3DSSG dataset, which includes 160 object labels and 26 predicate labels. This comprises diverse information collected from 1,553 real-world indoor scenes, including point clouds, text annotations, and RGBD sequences.
> > > - For Open3DSG [C], evaluation was performed using zero-shot inference with CLIP and LLM on the 3DSSG validation split.
> > >
> > > Additionally, the values in the table (e.g., **0.98/0.99**) from our **[Q2, W2]** response represent the **top-K Recall (R@K)** metrics of object, predicate and triplet from our model scaled to a **0-1 range**. Scene graph generation studies normalize metrics either to 0-1 or 0-100 ranges. While we presented values in the 0-100 range in our Manuscript for more precise comparison, other studies often use the 0-1 range due to various factors such as readability and table size considerations. Since Open3DSG reports values in the 0-1 range, we scaled the values from our model (some values are already presented in Manuscript **Table 2**) for ease of comparison. We apologize for any confusion caused by not explicitly mentioning this scaling.

---

> > > > ### Comment · Reviewer_dCsV · 2025-08-01
> > > >
> > > > Thanks for the responses from authors. I will adjust my score in the final discussion period accordingly.

---

> > > > > ### Author Response · Authors · 2025-08-01
> > > > >
> > > > > We sincerely appreciate the reviewer's adjustment of the score and will do our best to incorporate the feedback to create a more complete paper.

---

### Official Review · Reviewer_KTsC · 2025-06-26

**Clarity:** 2
**Significance:** 2
**Originality:** 2
**Rating:** 5
**Confidence:** 3

**Summary:**

This paper tackles the supervised 3D semantic scene graph prediction task. The authors begin with a set of preliminary observations highlighting a bottleneck in prior work: misclassification of object categories often degrades the overall graph prediction performance. To address this, they propose a discriminative learning objective powered by CLIP for the feature encoder to improve object classification.
In addition, they identify a gap in prior work, which is the lack of leveraging detailed modeling of relationship information of graph data in the training step. To this end, the authors introduce several GNN-based modules aimed at enhancing edge features: 1) Local Spatial Enhancement (LSE) enforces the reconstruction of embedded edge features back into raw relational features to strengthen understanding, 2) Bidirectional Edge Gating (BEG) learns distinct edge embeddings depending on directionality, and 3) Global Spatial Enhancement (GSE) explicitly captures both local and global relationships based on node distance.
Together, the improvements in discriminative object feature learning and enhanced GNN edge modeling contribute to performance gains over state-of-the-art methods.

**Questions:**

1. Regarding W3: Would it be possible to tweak the current method and test it under self-supervised or weakly supervised settings?
2. Regarding W1: Could the authors further clarify how the OFL module differs or is unique from existing 3D cross-modal pipelines?  Specifically, how does this design contribute uniquely to the scene graph prediction task?
3. It would be interesting to revisit the trends (Table 1 and Fig. 2) identified in the preliminary observations after applying the proposed method. Do the improvements directly address the issues noted earlier?
4. The Global Spatial Enhancement module seems functionally similar to mechanisms in established architectures like GATs [a], which also aggregate context using learned attention weights. Could the authors clarify how GSE differs and why it was preferred over attention-based alternatives? (Notably, the paper does not compare different GNN backbones.)

[a] Veličković, Petar, et al. "Graph Attention Networks." ICLR. 2018.

**Ethical Concerns:**

["NO or VERY MINOR ethics concerns only"]

**Final Justification:**

The authors have addressed most of my concerns and questions. I believe their additional experiments, both for my points and for other reviewers’, can add clear value to the paper. Especially, the work’s improved relevance to the current emerging research direction with VLM/LLM advances (as raised by Reviewer dCsV) and my point about recent works in weakly supervised settings are well supported. I originally leaned toward acceptance, and with these additions, I would like to increase my score to weak accept.

**Limitations:**

The authors do mention certain limitations. Additional potential concerns have been discussed above.

**Paper Formatting Concerns:**

No issue

**Quality:**

3

**Strengths And Weaknesses:**

### Strengths
1. The authors provide source code, which helps ensure the full reproducibility of their results.
2. The paper presents a clear motivation, starting from preliminary observations and theoretical insights. The narrative is also coherent and technically sound.
3. Comprehensive experimental results and ablation studies are included.
4. The proposed method achieves strong performance and outperforms recent works in the same supervised setting.

### Weaknesses
1. The technical novelty is somewhat limited. While the model achieves strong results, much of the improvement appears to come from the OFL (Object Feature Learning) module, whose standalone use (Table 5, row 1) already surpasses baselines like SGFN and VL-SAT. However, OFL itself resembles standard cross-modal feature learning, which is already well-established in the field (e.g., [30] also learns joint features across 2D, text, and point clouds).
2. The additional GNN components (LSE, BEG, and GSE) yield only marginal gains and may not justify the added complexity.
3. The latest direct competitor in the same fully supervised setting is CVPR 2023 [31]. Recent work from 2024–2025 shifts toward self-supervised or weakly supervised setups [30], and it would be insightful to evaluate how the proposed method compares under those recent settings.
4. Some mathematical notations are unclear or incomplete. For example, $\bar{z}$ (L-231) is never properly defined, and it is unclear how the weighted edge in Eq. (6) relates to the BEG module.

---

> ### Author Rebuttal · Authors · 2025-07-30
>
> We sincerely appreciate reviewer's thoughtful review and have made every effort to address reviewer's concerns.
>
> # [W1, Q2] Contribution of OFL
> While existing studies [30, 31] that utilize joint features from text, 2D, and point cloud data actively leverage RGBD images and text information, they overlook a critical aspect we identified in **Section 2 (Observation)**: object classification accuracy and entropy directly impact predicate prediction performance. This key insight drove us to design a specialized approach for scene graph prediction.
>
> To address this overlooked factor, we adopted a cross-modal setting to learn more discriminative object representations. While this aligns with other 3D object representation studies [1, 39], our approach differs significantly in implementation. Specifically, we employed **supervised contrastive learning** but intentionally removed the **positive denominator** from the loss function, enabling extremely low-entropy classification by pushing same-class point clouds closer while separating different classes further (experimental results for this are provided in **Appendix Table 2**). Furthermore, we leveraged **multi-view** RGBD images for contrastive learning, maximizing the discriminative power of learned representations.
>
> These design choices – the modified supervised contrastive loss and multi-view RGBD utilization – contribute uniquely to the scene graph prediction task, resulting in substantial performance improvements through OFL (please refer to **Table 4** in the Manuscript). To our best knowledge, OFL is the first 3D cross-modal pipeline designed specifically for scene graph prediction's unique characteristics, representing a task-specific innovation rather than merely applying existing methods.
>
> # [W2] Cost-Effective Gains of GNN components
> As shown in **Table 3 of the Appendix**, GSE/LSE/BEG make clear contributions to performance improvement. When GSE is applied, SGCls **R@K** and **mR@K** improve by **2-3%**, with overall performance gains across all metrics.
>
> LSE and BEG show much greater performance gains when combined with GSE compared to their individual use. Notably, BEG with GSE improves Predicate, SGCls, and PredCls **mR@K** by **3-6%**, a significant improvement considering the 3DSSG dataset's long-tailed distribution.
>
> These improvements are possible because each module captures different aspects of scene context that cannot be captured by object features alone. GSE captures global context while LSE and BEG capture local relational context, working complementarily.
>
> More remarkably, these gains are achieved efficiently. As shown in the table below, all three modules combined constitute only about **16%** of total training parameters. LSE achieves meaningful improvements with just **0.29%** of parameters, while even the largest module, GSE, uses less than **10%**. This demonstrates that our proposed modules achieve substantial performance improvements with minimal computational overhead. We appreciate the reviewer pointing out this missing aspect, and we will add these complexity-related experiments to the Manuscript.
>
> ||||
> |---|---|---|
> |Module|# of training parameters|% of total training parameters|
> |GSE|2.1M (2,103,296)|9.3%|
> |BEG|1.4M (1,445,377)|6.4%|
> |LSE|67K (67,361)|0.29%|
>
> Number of total training parameters: 22.5M
>
> # [W3, Q1] Extension to the weakly-supervised settings
> We tested our approach under the weakly-supervised setting from Wang et al. [30]. Given the rebuttal period, we maintained all our proposed components (OFL, GSE, LSE, BEG) while incorporating only their weakly-supervised learning methodology with pseudo-label.
>
> |Method|SGCls||||PredCls||||
> |---|---|---|---|---|---|---|---|---|
> ||R@20|mR@20|R@50|mR@50|R@20|mR@20|R@50|mR@50|
> |Wang et al. [30]|18.7|11.3|21.8|13.9|53.5|31.8|64.4|41.2|
> |Ours + weakly-supervised|25.5|18.1|28.2|21.2|61.5|38.8|69.7|45.4|
> |Ours|**36.1**|**35.3**|**37.7**|**37.7**|**70.2**|**58.8**|**82.0**|**66.1**|
>
> As shown in the table above, our method with weakly-supervised settings achieves better **R@K** and **mR@K** metrics than Wang et al. [30] across all tasks, though performance is lower than our fully-supervised approach. This is expected due to the inherent noise in weakly-supervised signals with inaccurate triplet labels.
>
> Importantly, this experiment validates that our insights remain effective even under weakly-supervised conditions, demonstrating the extensibility of our approach. We believe this represents a simple yet efficient advancement that can enhance performance across various 3D Scene Graph Generation settings. We will also release the code for this experiment upon publication.
>
> # [Q3] Experiments on preliminary observations
> An important point to clarify is that the trends presented in **Table 1** and **Figure 2** of our observation section are not **"problems"** to be solved, but rather **"phenomena"** we discovered in the scene graph generation task. Notably, they reveal how object classification errors propagate and influence predicate prediction—a fundamental characteristic of the task. Understanding and leveraging this dependency forms the foundation of our proposed approach. The table below compares predicate errors according to object and subject classification status between our model and VL-SAT [31].
>
> |||||
> |---|---|---|---|
> | |Obj. ✓|Obj. ✓/✗|Obj. ✗|
> |model|Sub. ✓|Sub. ✗/✓|Sub. ✗|
> |VL-SAT|8.0%|12.5%|19.1%|
> |Ours|7.5%|11.5%|18.2%|
>
> This shows that our model also exhibits the pattern where predicate error rate increases with object error rate, demonstrating that the fundamental phenomenon discovered in our observation persists.
>
> Notably, the effectiveness of our proposed object encoder is particularly remarkable. As shown in **Table 2** of the Manuscript, Object **R@1** improved by **3.6%** compared to VL-SAT, increasing the number of correctly predicted triplets. More importantly, despite improved object prediction accuracy, the predicate error rate actually decreased. If our hypothesis does not worked, we would expect the predicate error moves different from that of others or error increases. However, the results align precisely with our hypothesis: more discriminative object representations lead to more robust predicate predictions, demonstrating that the predicate estimator can be trained much more robustly.
>
> Furthermore, the lower predicate error rates across all categories compared to VL-SAT suggest that our GNN components (GSE, LSE, BEG) effectively utilize scene context to improve predicate prediction.
>
> These results demonstrate that we have understood and leveraged the phenomena discovered in our observation to improve overall performance. The trends we observed reflect intrinsic characteristics of scene graph generation and can provide valuable insights for future work.
>
> # [Q4] Difference between the attention mechanisms of GSE and GAT
> The attention mechanisms in GSE and GAT, while both utilizing attention, have fundamental differences in their design and purpose.
>
> **GSE's Distance-based Attention**: GSE explicitly integrates **physical distances** between objects into attention computation. This incorporates geometric context of the scene that cannot be expressed by individual object features alone, enabling the model to effectively learn scene context before GNN input (**Equation (6)** of the Manuscript).
>
> **GAT's Feature-based Attention**: In contrast, GAT's attention mechanism calculates attention based solely on **similarity between node features**. It relies purely on feature representation similarity without considering explicit spatial information such as physical distance.
>
> The key difference is that GSE employs a hybrid approach combining feature similarity with explicit spatial distance information, while GAT relies solely on feature similarity for attention computation.
>
> To verify GSE's effectiveness beyond GAT backbone, we conducted additional experiments using GCN as the backbone.
>
> |||||||||
> |---|---|---|---|---|---|---|---|
> |||Object||Predicate||Triplet||
> |GSE|Backbone|R@1|R@5|R@1|R@3|R@50|R@100|
> ||GCN|58.41|80.92|83.96|96.24|89.69|92.34|
> |**✓**|GCN|**59.70**|**81.07**|**84.78**|**96.70**|**90.26**|**92.76**|
>
> As shown in the table above, GSE's effectiveness remains clear even with GCN backbone. This demonstrates that GSE's distance-based attention operates independently from GAT's feature-based attention.
>
> # [W4] Incomplete notations
> ### **Regarding the definition of $\bar{\mathbf{z}}$**
> $\bar{\mathbf{z}}$ is the refined node feature after GSE processing, as mentioned in L-224. However, as reviewer correctly pointed out, this definition appears before rather than after Equation (7), causing confusion. To improve clarity, we will revise the text to explicitly define "$\bar{\mathbf{z}}$ as the refined node feature with spatial context incorporated through the GSE module" immediately after Equation (7).
>
> ### **Relationship between Equation (6) and BEG module**
> We apologize for the confusion. The weighted edges in Equation (6) are not directly related to the BEG module. Equation (6) describes the distance-based attention mechanism of the GSE module, which is used solely for updating object features. The representation in Figure 4 showing GSE updating edge features may have caused this misunderstanding. We will revise Figure 4 to reflect this clear distinction.
>
> ### **Notation improvements**
> We will remove and unify redundant notations such as $\bar{\mathbf{z}}\_i$ and $\mathbf{f}\_i^{node}$, $\tilde{\mathbf{z}}\_{ij}^e$ and $\mathbf{f}\_{ij}^{edge}$. Additionally, we will separate the inline formulas related to node and edge updates into separate equations to improve readability:
>
> $\bar{\mathbf{z}}\_i \leftarrow \mathrm{LN}(\mathrm{MLP}(\mathrm{CAT}(\bar{\mathbf{z}}\_i, \sigma(a\_i))))$
>
> $\mathbf{z}^e\_{ij} \leftarrow \mathrm{MLP}\left(\mathrm{CAT}\left(\bar{\mathbf{z}}\_i, \mathbf{z}^{e}\_{ij}, \tilde{\mathbf{z}}^{e}\_{ij}, \bar{\mathbf{z}}\_j\right)\right)$

---

> > ### Comment · Reviewer_KTsC · 2025-08-02
> >
> > Thank you to the authors for taking the time to provide thoughtful and detailed responses to all reviewers!
> > The motivations and justifications behind their proposed object feature learning, as well as other design choices, were clearly addressed in my Q2 and Q4. Their response to Q4 also strengthens the paper’s relevance by demonstrating significant improvements (10-20\%) over recent (2024) methods in weakly supervised settings.
> > My main concerns have been addressed, and I am willing to increase my score by one point.
> > However, I agree with the other reviewers about the difficulty of following the method section. I encourage the authors to revise this in the final version by incorporating the clarifications and improvements promised in the rebuttal.

---

> > > ### Author Response · Authors · 2025-08-02
> > >
> > > We are glad that our rebuttal has addressed the reviewer's concerns. We sincerely thank the reviewer for considering to increase the score. Regarding the method section, as promised in our rebuttal, we will make every effort to clarify it and incorporate these improvements into the final version. Once again, we thank the reviewer for the valuable feedback.

---

### Official Review · Reviewer_YmTJ · 2025-07-03

**Clarity:** 2
**Significance:** 2
**Originality:** 3
**Rating:** 4
**Confidence:** 3

**Summary:**

This paper focuses on 3D Semantic Scene Graph (3DSSG) prediction, addressing limitations in existing methods such as object misclassification propagating to relationship errors and insufficient integration of object information when constructing relationship features. It proposes a discriminative object feature encoder with contrastive pretraining, a novel relationship feature encoder combining semantic and geometric features, and a GNN with global spatial enhancement and bidirectional edge gating. Experiments on the 3DSSG dataset show it outperforms state-of-the-art methods, with its object encoder improving other models when plugged in.

**Questions:**

1. Could you provide a detailed explanation of Distance-based Attention in the Global Spatial Enhancement mechanism
2. What is the computational complexity (e.g., FLOPs, training time) of the proposed method compared to baselines?
3. In the ablation experiments, why does the accuracy of triplet inference for the complete model perform worse than that of the model containing only the GSE module?

**Ethical Concerns:**

["NO or VERY MINOR ethics concerns only"]

**Final Justification:**

The authors' reply addressed my questions, and after considering their rebuttal and relevant discussions, I maintain my original score.

**Limitations:**

Yes.

**Paper Formatting Concerns:**

None.

**Quality:**

2

**Strengths And Weaknesses:**

Strengths:
1. The paper provides a detailed analysis of the problem and, leveraging statistical evidence, identifies the discriminability of object features as a critical bottleneck in 3DSSG prediction.
2. The discriminative object encoder effectively enhances object representation by leveraging cross-modal contrastive pretraining to align 3D, 2D, and text features.

Weaknesses:
1. The paper's title is inconsistent with its content. While the title references "object-centric representation learning," this approach is neither employed in the paper nor is the term "object-centric" itself ever used.
2. The details in the method section lack clarity, particularly regarding the Global Spatial Enhancement mechanism, which impedes a comprehensive understanding of the approach.

---

> ### Author Rebuttal · Authors · 2025-07-29
>
> We are grateful for reviewer's insightful review and have made every effort to address the concerns reviewer have highlighted.
>
> # [W1] Title–Content Alignment
> We completely agree with reviewer's observation regarding the inconsistency between the title and content. We acknowledge that while we used the term "object-centric representation learning" in our paper title, we did not directly employ this terminology in the paper, instead using expressions such as "object feature learning" or "object representation." We apologize for any confusion this may have caused. To address this issue, we are considering revising the title to more accurately reflect the actual content and contributions of our paper:
>
> - **Object Feature Learning for Enhanced 3D Semantic Scene Graph Prediction**
>
> This title more clearly and precisely represents the research we actually conducted–improving scene graph generation performance through enhanced discrimination power of object features. By adopting this revised title, we will unify the terminology throughout the paper as Object Feature Learning (OFL), thereby ensuring consistency between the title and content.
>
> We are grateful once again for reviewer's valuable feedback and will work to improve the clarity of our paper by enhancing the consistency of our title and terminology.
>
> # [W2, Q1] Distance-Based Attention in GSE
> Distance-based Attention serves as a core component of the GSE (Global Spatial Enhancement) module, integrating distance information between object instances into object features through a self-attention mechanism. This approach incorporates the geometric context of the scene, which cannot be expressed by individual object features alone, prior to GNN input, thereby enabling the model to effectively learn scene context.
>
> ### **Mechanism of Distance-based Attention**
> The specific operation can be described as follows. Given the distance $d_{ij}$ between object instance pair $(i,j)$ in the scene, we can represent the distances of all object pairs as a matrix $ D=[d_{ij}]_{i,j = 1,\ldots,N}$. This distance matrix is multiplied by learnable parameters $W^{(h)}\in\mathbb{R}^{N \times N}$ to generate distance weights $w\_{ij}^{(h)}=W^{(h)}D$. The computed distance weights are then integrated into the standard attention mechanism as follows (**Equation 6** in the Manuscript):
> $\alpha\_{ij}^{(h)} = \text{softmax}\_j \left( \frac{\mathbf{q}\_i^{(h)\top}\mathbf{k}\_j^{(h)}}{\sqrt{d\_k}} + \mathbf{w}\_{ij}^{(h)} \right)$
>
> ### **Benefits and Performance Impact**
> This distance-based attention globally reorganizes the spatial relationships of objects within the scene, emphasizing geometrically meaningful relationships while effectively filtering object pairs with low relevance. Attention between spatially proximate objects can be strengthened, or object pairs with specific distance patterns can be emphasized, enabling the model to better understand the structural characteristics of the scene.
>
> As confirmed in the second row of **Table 5** in the Manuscript, this mechanism brought substantial performance improvements. Through this approach, the model can generate more accurate scene graphs by comprehensively considering not only the individual characteristics of objects but also their spatial arrangement and context within the scene.
>
> We appreciate the reviewer for pointing out the insufficient explanation of the GSE module, and we will strengthen this content in the Manuscript.
>
> # [Q2] Computational Complexity of the proposed method
> The following table compares the training time, FLOPs, and total number of training parameters between the two models:
>
> |  |  |  |  |
> | --- | --- | --- | --- |
> | Model | Training Time* | FLOPs | # of training parameters |
> | VL-SAT | 37 hours (in NVIDIA RTX 3090) | 156M (156,618,160) | 27.1M (27,162,021) |
> | Ours | 45 hours (in NVIDIA RTX 3090) | 189M (189,405,982) | 22.5M (22,564,199) |
>
> *Training Time: Running time for 100 epochs.
>
> As shown in the table, our model has slightly fewer parameters compared to VL-SAT, but marginally higher FLOPs. This difference stems from the following reasons:
>
> **FLOPs**: Our proposed BEG (Bidirectional Edge Gating) and GSE (Global Spatial Enhancement) modules require additional computational operations.
>
> **Training Parameters**: In our model, the object encoder uses pretrained weights and does not require separate training. This is a major factor in reducing the total number of trainable parameters.
>
> Consequently, there is a trade-off where training time becomes somewhat longer due to increased FLOPs, but we believe this is a reasonable cost for performance improvement. Specifically, compared to VL-SAT, our method achieves substantial gains across all metrics - for instance, **Object R@1** improves from 55.93% to **59.53%**, and **PredCls (w/ GC) R@50** increases from 79.9% to **82.0%**, demonstrating that the additional computational cost yields significant performance benefits. It is particularly noteworthy that we effectively reduced training parameters by utilizing a pretrained object encoder.
>
> We appreciate reviewer's observation regarding this aspect we had overlooked, and we will add this content to the Appendix.
>
> # [Q3] Analysis of the ablation study
> As shown in **Table 5** of the Manuscript, the model with only the GSE module demonstrates 0.1% higher performance in Triplet R@50 compared to the model with all modules. This can be reasonably explained by considering the characteristics of each module.
>
> ### **GSE Module Characteristics**
> GSE captures global spatial information within the scene to learn the overall context of object arrangement. Specifically, GSE applies attention mechanisms across all object features within the scene to understand global context. This global attention approach inevitably assigns greater weights to frequently appearing objects and their relationships in the dataset. Consequently, GSE can effectively improve prediction accuracy for frequently occurring head predicates. While this is highly effective for enhancing performance on dominant classes, it shows minimal effect in alleviating class imbalance problems.
>
> ### **LSE and BEG Module Characteristics**
> In contrast, LSE and BEG focus on learning local spatial information between object pairs. These modules capture unique spatial patterns of individual relationships, modeling local features that GSE might miss. Therefore, they play crucial roles in learning spatial characteristics of less frequent predicates.
>
> ### **Module Interaction and Performance**
> When LSE and BEG are additionally introduced, the model allocates more attention to local spatial patterns. During this process, the global bias toward head predicates is weakened, resulting in a slight decrease (0.1%) in R@50, but simultaneously the prediction capability for body/tail predicates is significantly improved, leading to approximately **3%** increase in **mR@50 (Top-K mean Recall)**. The important point is that adding LSE and BEG barely impairs GSE's ability to understand global context. Rather, these modules work complementarily, adding local details on top of the global understanding provided by GSE, thereby improving the model's overall generalization power.
>
> To summarize, by combining GSE, LSE and BEG, the model has moved closer to learning unique spatial characteristics of each relationship, thereby removing bias from long-tailed distribution. Consequently, the model using all modules performs more balanced predictions and demonstrates better generalization performance.

---

> > ### Comment · Reviewer_YmTJ · 2025-08-05
> >
> > Thank you for your response, which has addressed my questions. Please revise the final version in accordance with the promised clarifications and improvements in your reply.

---

> > > ### Author Response · Authors · 2025-08-05
> > >
> > > We sincerely appreciate for your time to respond to our rebuttal. As you suggested, we will make sure to incorporate the points mentioned in our rebuttal into the final version.

---

### Official Review · Reviewer_eReq · 2025-07-05

**Clarity:** 3
**Significance:** 3
**Originality:** 2
**Rating:** 4
**Confidence:** 3

**Summary:**

This paper addresses 3D Semantic Scene Graph Prediction by systematically analyzing the impact of cascading errors, particularly the effect of object misclassification on relational inference. The authors propose to decouple object representation learning from scene graph prediction, employing a supervised contrastive learning objective to enhance object feature discriminability. Furthermore, the method integrates both global and local spatial information to better capture object relations. Extensive experiments and ablations on the 3DSSG benchmark demonstrate significant improvements over three strong baselines, validating the effectiveness of the proposed approach and its key components.

**Questions:**

1. Are the reported results statistically significant? Given known variations across different random seeds, providing an empirical explanation or conducting experiments with statistics over multiple runs would strengthen the results.
2. How do object representation and scene graph modeling individually contribute to final performance? Is the proposed scene graph method necessary, or would strong object representations alone suffice?
3. Table 2 in the Appendix suggests that aligning object representations to CLIP text embeddings is crucial. Can the authors provide empirical or intuitive explanations for this, and are there any observed limitations when using CLIP visual modality?

**Ethical Concerns:**

["NO or VERY MINOR ethics concerns only"]

**Final Justification:**

The authors’ detailed response has successfully resolved my earlier questions and concerns. The new experimental evidence and discussion further strengthen the paper. I believe the work now meets the threshold for acceptance. I will keep my score at 4, which is positive towards acceptance.

**Limitations:**

yes

**Paper Formatting Concerns:**

Table captions should appear above the tables, as per conference guidelines.

**Quality:**

3

**Strengths And Weaknesses:**

Strengths:

1. The paper provides a thorough analysis of cascading errors in 3D semantic scene graph prediction, identifying object representation as a key bottleneck—an important and insightful contribution.
2. Experimental quality is solid, with comparisons to three baselines on the 3DSSG dataset and comprehensive ablation studies on key components.
3. The paper is clearly written, making it easy to read and follow.

Weaknesses:

1. The use of CLIP features for 3D tasks has been explored in recent works [1][2], which limits the novelty of this aspect.
2. The contributions are somewhat split: while the analysis convincingly shows the importance of object representation, the subsequent scene graph modeling components (Sections 3.2 and 3.3) appear more ad-hoc and less central.
3. Table 2 appears incremental, and lacks error bars or statistics over multiple seeds, making it difficult to assess the statistical significance of the results.

[1] Chen, Runnan, et al. "Clip2scene: Towards label-efficient 3d scene understanding by clip." CVPR 2023.
[2] Hegde, Deepti, Jeya Maria Jose Valanarasu, and Vishal Patel. "Clip goes 3d: Leveraging prompt tuning for language grounded 3d recognition." CVPR 2023.

---

> ### Author Rebuttal · Authors · 2025-07-29
>
> We sincerely appreciate reviewer's thoughtful review and have endeavored to address the concerns reviewer have raised to the best of our ability.
>
> # [W1] Usage of CLIP features
> While we acknowledge that research utilizing CLIP features for 3D tasks already exists as reviewer mentioned, we would like to clarify that our research approach and objectives are fundamentally different from existing studies.
> ### **Distinctions from Existing Research**
> The existing studies reviewer referenced [A,B], along with other related works [C,D], primarily aim to transfer CLIP's powerful generalization capabilities to the 3D domain, focusing on zero-shot or few-shot learning performance. These studies concentrate on developing open-vocabulary 3D recognition models applicable to various downstream tasks (semantic segmentation, 3D object classification, retrieval, etc.) without costly human annotation.
>
> In contrast, our work operates in a **closed-vocabulary setting**, where our core objective is to validate the hypothesis presented in **Section 2** of our Manuscript: that predicate classification errors are strongly correlated with incorrect or uncertain object label predictions. To achieve this goal, we adopted CLIP-based **supervised contrastive learning** to effectively utilize the RGBD images and textual (label) information in the 3DSSG dataset. Rather than simply using CLIP as a source for knowledge distillation, we employed it as a tool to accomplish our specific research objectives.
>
> ### **Novel Contributions of Our Research**
> The core novelty of our proposed method lies in the newly designed loss function. Specifically, we intentionally **removed the positive denominator** from the conventional supervised contrastive loss.  As shown in **Appendix Table 1**, this design achieved higher accuracy than existing methods by removing the pressure to reduce intra-class distances, allowing the model to focus on inter-class separation and enhance discrimination power. Furthermore, compared to other studies [A, B, C, D], we leveraged multi-view RGBD images as positive sample of supervised contrastive scheme. We believe that proposed loss function can align semantic cues to 3D features more effectively by using multi-view information. This approach aligns with our core hypothesis and is supported by both the mathematical analysis in **Appendix Section 2.2** and our experimental results. To our best knowledge, this is the first attempt at such loss modification in 3D object classification tasks using supervised contrastive learning. Furthermore, when combined with our scene graph methods (GSE, LSE, BEG), these highly discriminative object features enable more accurate relationship reasoning, creating a synergistic effect that significantly improves overall scene graph generation performance.
>
> [A] Chen et al. "Clip2scene: Towards label-efficient 3d scene understanding by clip." CVPR 2023.
>
> [B] Hegde et al. "Clip goes 3d: Leveraging prompt tuning for language grounded 3d recognition." ICCV 2023.
>
> [C] Afham et al. "Crosspoint: Self-supervised cross-modal contrastive learning for 3d point cloud understanding." CVPR 2022.
>
> [D] Zeng et al. "Clip2: Contrastive language-image-point pretraining from real-world point cloud data." CVPR 2023.
>
> # [W2, Q2] Necessity of scene graph methods
> While our proposed object representation methodology (OFL) has brought substantial performance improvements on its own, additional scene graph methods are necessary to maximize these effects and optimize overall scene graph prediction performance. Since object features inherently capture only individual object characteristics without spatial relationships or positional information between objects, our proposed modules enable effective utilization of such scene context information through the following specific roles:
>
> - **GSE** incorporates distance information between all objects in the scene into each object feature through self-attention. This enables comprehensive learning of the overall object arrangement context within the scene.
> - **LSE** learns geometric information (distance differences, size differences) between two object instances within a specific relationship as an auxiliary task. This helps the relationship encoder effectively utilize local spatial information.
> - **BEG** reflects the principle that relationships should vary depending on whether an object's role is subject or object, enabling relationship modeling that considers symmetry.
>
> **Table 5** of the Manuscript demonstrates the effectiveness of our scene graph methods, where the topmost case where all three modules are not applied represents the results of applying only our object representation methodology, OFL. Compared to this baseline, the model with all three modules applied showed substantial improvements in **Top-K mean Recall(mR@K)** performance, with particularly notable gains of approximately **6%** in both Predicate **mR@1** and PredCls **mR@50**. Given the long-tail distribution challenges (class imbalance problem) inherent in 3DSSG predicate labels, these improvements indicate that our modules significantly enhance the model's generalization capability.
>
> Additionally, the table below compares scene graph prediction results with and without scene graph methods (Additional results will be provided in the Appendix). While OFL alone demonstrates strong object classification performance, it shows more frequent errors in predicting predicates between objects compared to the full model with scene graph methods.
> ||||||||||
> |---|---|---|---|---|---|---|---|---|
> ||**GT**|||**w/ scene graph methods**|||**w/o scene graph methods**||
> |Object|Predicate|Subject|**l** Object|Predicate|Subject|**l** Object|Predicate|Subject|
> |tv|standing on|tv stand|**l** tv|standing on|tv stand|**l** tv|standing on|tv stand|
> |tv stand|standing on|floor|**l** tv stand|standing on|floor|**l** tv stand|standing on|floor|
> |tv stand|left|stool|**l** tv stand|left|stool|**l** tv stand|left|stool|
> |stool|right|tv stand|**l** stool|right|tv stand|**l** stool|right|tv stand|
> |stool|standing on|floor|**l** stool|standing on|floor|**l** stool |**closed by**|floor|
> |cabinet|attached to| wall| **l** cabinet |attached to| wall| **l** **tv**|attached to|wall|
> |plant|standing on|stool|**l** plant|standing on|stool|**l** plant|standing on|stool|
> |plant|none|floor|**l** plant|none|floor|**l** plant|**standing on**|floor|
> |plant|none|tv stand|**l** plant|none|tv stand|**l** plant|**left**|tv stand|
> |tv stand|none|plant|**l** tv stand|none|plant|**l** tv stand|**right**|plant|
>
> *Scene ID: 0cac7584-8d6f-2d13-8df8-c05e4307b418
>
> In conclusion, the proposed GSE, LSE, and BEG modules were designed to organically combine with the OFL methodology to maximize the effects of improved object representation. This integrated approach effectively combines object-level improvements with scene-level contextual information to achieve overall performance enhancement, which we believe constitutes another important contribution of our research.
>
> # [W3, Q1] Experiment conducted with different random seeds
> ||||||||
> |---|---|---|---|---|---|---|
> ||Object||Predicate||Triplet||
> |Model|R@1|R@5|R@1|R@3|R@50|R@100|
> |Ours|59.97±0.41|81.18±0.11|91.11±0.28|98.51±0.05|91.39±0.19|93.69±0.18|
> ||||||||
>
> ||||||||||||||
> |---|---|---|---|---|---|---|---|---|---|---|---|---|
> | | |SGCls (w/ GC)| | |PredCls (w/ GC)| | |SGCls (w/o GC)| | |PredCls (w/o GC)| |
> |Model|R@20|R@50|R@100|R@20|R@50|R@100|R@20|R@50|R@100|R@20|R@50|R@100|
> |Ours|36.3±0.25|38.1±0.36|38.2±0.36|71.0±0.40|82.0±0.33|82.6±0.35|38.7±0.41|47.5±0.45|52.9±0.39 |73.6±0.42|87.8±0.29|94.5±0.30|
> ||||||||||||||
>
> To address concerns about statistical significance, we conducted experiments using **8 different random seeds (2029-2036)** and present the mean and standard deviation of the results in the table above.
>
> As shown, all evaluation metrics exhibit very low standard deviations (mostly below **0.5**) across multiple runs, indicating that our method achieves consistent and stable performance regardless of random initialization. This low variability demonstrates that the reported performance improvements are robust and reproducible, not dependent on specific random seeds, thereby strengthening the validity of our findings. We will transparently release all code for these experiments to ensure reproducibility.
>
> # [Q3] Primacy of CLIP Text Modality
> As shown in **Appendix Table 2**, we believe that CLIP's text embedding plays a more important role than visual embedding due to the characteristics of the RGBD images provided by the 3DSSG dataset.
>
> **Challenges with Visual Modality**: The 3DSSG dataset contains RGBD image sequences captured directly from real scenes, which present several challenges for visual feature extraction. Visual noise exists because multiple objects other than the target object are captured together in the same image. Additionally, some images are blurry or unclear, creating difficulties in extracting reliable visual features.
>
> **Advantages of Text Modality**: In contrast, text embeddings are generated through a standardized prompt "A point cloud of {LABEL}", providing consistent and clear representations for each object class. This difference explains the significant performance degradation observed in **Appendix Table 2** when the text modality is excluded.
>
> However, these results do not indicate fundamental limitations of methodologies utilizing CLIP visual modality. Rather, our approach effectively leverages the complementary characteristics of both modalities. The visual modality captures rich semantic and geometric characteristics of objects through multi-view images, while the text modality provides clear and consistent semantic anchors. This multi-modal design ensures robustness across varying data quality, not limited to 3DSSG. In environments with higher-quality image data, we anticipate the visual modality's contribution would increase, yielding further performance improvements.

---

### Note · Authors · 2025-08-12

We sincerely thank the Area Chair and all Reviewers for dedicating their time and effort to evaluating our work. We have made our best efforts to address all concerns raised by the reviewers, and we are deeply grateful that all reviewers have confirmed their concerns have been resolved, which has helped us significantly improve our paper. We will further enhance the quality of our manuscript by faithfully incorporating the following revision suggestions:

**Reviewer eReq [W2, Q2]:** We will add qualitative experiment results demonstrating the efficiency of our scene graph module to the appendix.

**Reviewer YmTJ [W1]:** We will change the paper title to "Object Feature Learning for Enhanced 3D Semantic Scene Graph Prediction."

**Reviewer YmTJ [W2, Q1]:** Following the feedback that the GSE module in Section 3.3 is difficult to understand, we will add more detailed explanations as discussed in the rebuttal.

**Reviewer YmTJ [Q2]:** We will include computational complexity analysis of our proposed method in the appendix.

**Reviewer KTsC [W2]:** We will add complexity analysis for each GNN component method to better highlight the superiority of our proposed modules.

**Reviewer KTsC [W3, Q1]:** We will add experiments applying our insights to weakly-supervised settings in the appendix to better demonstrate the extensibility of our research.

**Reviewer dCsV [W1, Q1]:** We will update the related works section to include recent 3D Scene Graph Generation studies utilizing VLM/LLM, reflecting the latest research trends.

**Reviewers KTsC and dCsV:** We will revise the BEG and GSE sections (Section 3.3) as specified in the rebuttal to improve readability.

We once again thank all reviewers for their constructive feedback that has helped elevate the quality of our work. We would also like to acknowledge **Reviewers KTsC and dCsV, who mentioned final score increments** upon resolution of all concerns.

---

### Decision · Program_Chairs · 2025-09-17

**Decision:**

Accept (poster)

**Comment:**

The final ratings for this paper are mixed (one "Accept," two "Borderline Accept," and one "Borderline Reject"), leaning toward acceptance though.
After a review of the paper, rebuttal, and discussions, the AC recommends acceptance.

The AC agrees with the reviewers that the paper's core contribution of identifying object representation quality as a key bottleneck in 3D Scene Graph prediction is reasonable. Initial reviews, however, raised valid concerns about the technical novelty, the clarity of the GNN components, and the relevance of a closed-vocabulary approach in an era of VLM/LLM-based methods.

The authors' rebuttal addressed most of these concerns with new experiments and clarifications. They demonstrated the statistical robustness of their results, showed the method's extensibility to a weakly-supervised setting, and provided a justification for their closed-vocabulary focus. This response satisfied three of the four reviewers.

While one reviewer maintained their "Borderline Reject" rating due to the paper's closed-set paradigm, the AC finds the authors' rationale reasonable within the current paper's scope, and believes the paper's insights are a valuable contribution. Therefore, the AC recommends acceptance.